# Low Rank Transformer for Multivariate Time Series Anomaly Detection and Localization

**Charalampos Shimillas**[1,2]**, Kleanthis Malialis**[1]**, Konstantinos Fokianos**[3]**, Marios M. Polycarpou**[1,2]

[1]KIOS Research and Innovation Center of Excellence, University of Cyprus, Nicosia, Cyprus
[2]Department of Electrical and Computer Engineering, University of Cyprus, Nicosia, Cyprus
[3]Department of Mathematics and Statistics, University of Cyprus, Nicosia, Cyprus

`{shimillas.charalampos, malialis.kleanthis, fokianos.konstantinos, polycarpou.marios}@ucy.ac.cy`

## Abstract

Multivariate time series (MTS) anomaly diagnosis, which encompasses both anomaly detection and localization, is critical for the safety and reliability of complex, large-scale real-world systems. The vast majority of existing anomaly diagnosis methods offer limited theoretical insights, especially for anomaly localization, which is a vital but largely unexplored area. The aim of this contribution is to study the learning process of a Transformer when applied to MTS by revealing connections to statistical time series methods. Based on these theoretical insights, we propose the Attention Low-Rank Transformer (ALoRa-T) model, which applies low-rank regularization to self-attention, and we introduce the Attention Low-Rank score, effectively capturing the temporal characteristics of anomalies. Finally, to enable anomaly localization, we propose the ALoRa-Loc method, a novel approach that associates anomalies to specific variables by quantifying interrelationships among time series. Extensive experiments and real data analysis show that the proposed methodology significantly outperforms state-of-the-art methods in both detection and localization tasks. Code is available at: https://github.com/CharisShimillas/ALoRa.

## 1 Introduction

Driven by the rapid growth of the Internet of Things (IoT), real-world systems have become more complex and vulnerable to faults. These anomalies frequently result in abnormal patterns for a stream of MTS. In this context, the diagnosis of anomalies in MTS is of great importance to ensure the reliability, safety, and efficiency of critical systems. Due to the scarcity of labeled data, anomaly diagnosis is commonly formulated as an unsupervised learning problem. It typically involves two key tasks: anomaly detection, which determines which timestamps are anomalous, and anomaly localization, which identifies the specific time series responsible for the detected anomalies.

MTS data exhibit complex dynamics, including temporal dependencies (relationships over time) and spatial dependencies (relationships across series). Effective anomaly diagnosis depends on reliably estimating these spatio-temporal dynamics. Deep learning models are widely applied to this task for their strong representation learning capabilities, with Transformer-based architectures shown to be especially effective in modeling the complex dynamics of MTS (Zerveas et al., 2021). However, the limited available theoretical insights into their decision process undermine reliability and trust in safety-critical settings, while also complicating anomaly localization. Practitioners often raise important questions regarding the learning process, the interrelationships within the data learned by the model, and the need for a localization method, since detection alone provides limited practical value in complex large-scale systems. As illustrated in Fig. 1, two of the main contributions of this work are to address these open questions, which have remained largely unanswered.

In addition, several existing detection methods are built around the point-adjustment evaluation strategy, whereby detecting any point within an anomalous segment suffices to consider the entire segment as an observed anomaly (Xu et al., 2022; Song et al., 2023; Shen et al., 2020). This approach often inflates performance metrics, since even random scoring methods can appear com-

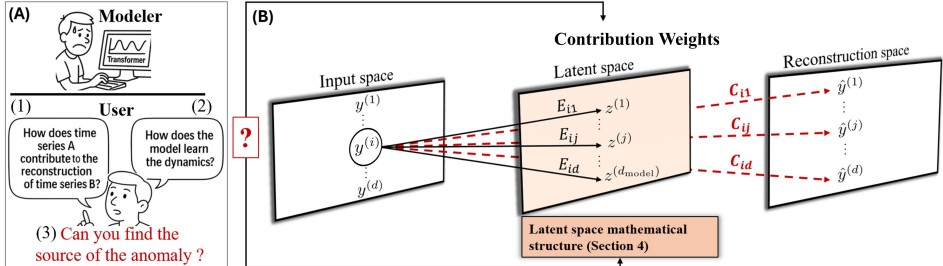

Figure 1: **(A):** Unresolved key questions commonly raised by practitioners. **(B):** The proposed localization method (Sec. 5.2), together with the theoretical analysis (Sec. 4), addresses these challenges. The figure illustrates the modeling process from the input space to the latent space and subsequently to the reconstructed outputs. An input series $i \in \{1, \cdots, d\}$ can influence each latent space representation and subsequently each reconstructed output. These relationships learned by the model are quantified through contribution weights: $E_{ij} \in \mathbb{R}$, capturing the influence of input series $i$ on latent feature $j \in \{1, \cdots, d_{\text{model}}\}$, and $C_{ij} \in \mathbb{R}$, capturing its influence on the reconstructed output series $j \in \{1, \cdots, d\}$. These weights enable effective localization and enhance interpretability.

petitive under such evaluation (Kim et al., 2022; Huet et al., 2022). To ensure reliable evaluation, trustworthy detection metrics have to accurately reflect the temporal characteristics of anomalies.

To address the challenges of reliable detection scoring, interpretability, and effective localization, this study makes the following contributions:

- **Theoretical insights of Transformer encoders on MTS:** We study how Transformer encoders learn from MTS data, shedding light on how the learned representations relate to classical time series. These insights advance the design of anomaly diagnosis methods and deepen the understanding of Transformers in sequential modeling. (Section 4)

- **Attention Low-Rank Transformer (ALoRa-T) for anomaly detection:** We propose the ALoRa-T method, which enhances anomaly detection by applying low-rank regularization to self-attention, and introduce a novel detection metric, the Self-Attention Low-Rank score (ALoRa-T score). This method outperforms state-of-the-art methods across multiple benchmark datasets, and captures effectively the temporal characteristics of anomalies. (Section 5.1)

- **ALoRa-Loc method for anomaly localization:** We introduce ALoRa-Loc, a localization method that first derives contribution weights that quantify the learned interrelationships between time series, offering key insights into the model's decision-making process. More critically, these weights capture how anomalies propagate across time series during modeling, enabling ALoRa-Loc to trace anomalies back to their origin and attribute them to the most relevant input time series. (Section 5.2)

## 2 RELATED WORK

**Anomaly detection:** Due to its importance for the safety and reliability of many critical infrastructure systems, such as water distribution networks (Eliades et al., 2025), power systems (Kyriakides & Polycarpou, 2015) and healthcare (Tang et al., 2022), unsupervised anomaly detection in MTS has attracted significant attention. Traditional unsupervised methods (Liu et al., 2008; Breunig et al., 2000; Schölkopf et al., 2001) have been applied to this task, but may not perform well since they do not learn complex dependencies. To address these limitations, deep learning models that capture temporal and inter-variable relationships have shown improvements over existing methods. A major class of deep learning approaches is reconstruction-based methods, typically using an encoder-decoder architecture trained in a self-supervised manner. The main idea is to learn meaningful latent representations that mimic the normal behavior. The models are expected to reconstruct normal sequences accurately. An unsuccessful reconstruction, in the sense of inflated errors, indicates potential anomalies in the data. Representative examples include RNN-based models such as

LSTM-VAE (Park et al., 2018), OmniAnomaly (Su et al., 2019), and InterFusion (Li et al., 2021), as well as the CNN-based model MSCRED (Zhang et al., 2019). More recently, Transformer models (Vaswani et al., 2017) have been proven highly effective for sequential modeling and well-suited to the complex dynamics of MTS (Nie et al., 2023). Anomaly-Transformer (Xu et al., 2022) introduces the AssDis metric, which compares each row of the self-attention matrix to a prior Gaussian distribution. MEMTO (Song et al., 2023) enhances a Transformer encoder with a gated memory module that learns memory items representing normal behavior. As new data arrives, it selectively updates these items. For anomaly detection, they introduce the LSD score, the distance between input queries and their closest memory items. Although the aforementioned models have shown strong performance under point-adjustment metrics, they may often not produce meaningful and reliable anomaly scores, as illustrated later in Figure 4. Lai et al. (2023) use the Performer model (Choromanski et al., 2021) to perform both point-wise and segment-wise reconstruction, defining the anomaly score as the ratio of their reconstruction errors. SARAD (Dai et al., 2024) uses a Transformer on transposed input windows, and thus the attention weights express feature-wise associations. To guide training and detect anomalies, SARAD relies on SAR metric, which tracks changes in attention weights, based on the intuition that anomalies disrupt feature associations.

**Anomaly Localization:** Anomaly localization has remained to a large extent an unexplored research area in deep learning. Recently it has begun to attract more attention, with some of the aforementioned approaches also addressing this critical issue. Localization effectiveness depends on two key factors: first, the model's ability to reconstruct normal patterns accurately. Second, the ability to interpret the reconstruction errors in a way that isolates the true sources of anomalies without being distorted by their effects. Some methods rely directly on the reconstruction error to inform the localization. For example, SARAD and OmniAnomaly (Su et al., 2019) perform localization by ranking the reconstruction errors of individual time series. While this provides a baseline approach, anomaly propagation can obscure true anomaly sources. More advanced techniques refine the reconstruction signal to improve localization. InterFusion (Li et al., 2021) applies Markov Chain Monte Carlo to adjust reconstruction errors. However, due to its computational complexity, it is limited to interpreting anomaly segments instead of individual time steps. DAEMON (Chen et al., 2023) leverages integrated gradients (Sundararajan et al., 2017) to attribute anomalies back to individual dimensions. Finally, AERCA (Han et al., 2025) focuses exclusively on the localization task, without addressing detection and localization simultaneously, while providing an interpretable approach and representing one of the most recent and effective methods in this area.

**Research gaps:** Although the aforementioned methods propose innovative architectures and scoring metrics, they often overlook key aspects: understanding what the model learns, how each time series is represented, and how these representations contribute to the final decisions. Such understanding is crucial for making detection applicable and trustworthy, and for enabling effective localization.

# 3 MULTIVARIATE TIME SERIES (MTS) NOTATION

A Multivariate Time Series (MTS) is a sequence of observations recorded over time across multiple variables. It is denoted as $\boldsymbol{Y} = [\boldsymbol{y}_1, \boldsymbol{y}_2, \ldots, \boldsymbol{y}_N]^\top$, where each row vector $\boldsymbol{y}_t \in \mathbb{R}^d$, $t = 1, \ldots, N$, denotes the multivariate observation at time step $t$. Each column vector of $\boldsymbol{Y}$, denoted as $\boldsymbol{y}^{(i)} \in \mathbb{R}^N$ represents the $i$-th univariate time series of length $N$, and $y_t^{(i)} \in \mathbb{R}$ is the value of the $i$-th series at time $t$. Given the streaming nature of time-series data, modeling the MTS at an arbitrary time $t$ relies on its history $\boldsymbol{Y}_{(t-T:t]}$, defined as the most recent $T$ observations up to $t$: $\boldsymbol{Y}_{(t-T:t]} = [\boldsymbol{y}_{t-T+1}, \ldots, \boldsymbol{y}_t]^\top \in \mathbb{R}^{T \times d}$. Applying this construction to the full MTS $\boldsymbol{Y} \in \mathbb{R}^{N \times d}$ produces a sequence of overlapping windows, each serving as the model input. Since the subsequent sections concern an arbitrary time $t$, for simplicity we denote the window $\boldsymbol{Y}_{(t-T:t]}$ by $\boldsymbol{Y}_{[t]}$.

# 4 THEORETICAL ANALYSIS OF TRANSFORMER ENCODER

This section presents a theoretical analysis of the learning dynamics of the Transformer encoder in the context of MTS. In particular, it relates the components of a standard Transformer model to classical statistical models.

**Embedding:** Given a MTS window at time $t$, denoted as $\mathbf{Y}_{[t]} \in \mathbb{R}^{T \times d}$, a Transformer encoder typically begins by embedding the input into a higher-dimensional space using a 1D-convolutional layer, i.e. it maps $\mathbf{Y}_{[t]}$ to $\tilde{\mathbf{Y}}_{[t]} \in \mathbb{R}^{T \times d_{\text{model}}}$, where $d_{\text{model}}$ is the embedding (model) dimension. Applying the 1D-convolution, the $k$-th embedded time series value at time step $t$, $\tilde{y}_t^{(k)}$, is computed as:

$$\tilde{y}_t^{(k)} = \sum_{i=1}^{d} \underbrace{\left( \sum_{j=-\frac{m-1}{2}}^{\frac{m-1}{2}} w_{i,j}^{(k)} \cdot y_{t+j}^{(i)} \right)}_{\text{Weighted average of the } i\text{'th raw time series}} \tag{1}$$

This operation is mathematically equivalent to a learnable **Vector Moving Average (VMA) filtering** (Brockwell & Davis, 2002), where the output at time $t$ is a weighted sum of multiple time series values in a local window around $t$. The filter weights $w_{i,j}^{(k)}$ determine the influence of time series $i$ on the output feature $k$, at lag $j$. The constant $m$ is the kernel size.

**Self-Attention Latent Space:** The output of the embedding layer, $\tilde{\mathbf{Y}}_{[t]} \in \mathbb{R}^{T \times d_{\text{model}}}$, is passed as input to the attention mechanism. At each layer $l$, the self-attention mechanism projects the previous layer's output $\boldsymbol{Z}^{(l-1)} \in \mathbb{R}^{T \times d_{\text{model}}}$ into the Query ($\boldsymbol{Q}^l$), Key ($\boldsymbol{K}^l$), and Value ($\boldsymbol{V}^l$) matrices using learnable projection matrices $\boldsymbol{W}^{(Q,l)} = \{w_{ij}^{(Q,l)}\}$, $\boldsymbol{W}^{(K,l)} = \{w_{ij}^{(K,l)}\}$, and $\boldsymbol{W}^{(V,l)} = \{w_{ij}^{(V,l)}\}$, each of dimension $\mathbb{R}^{d_{\text{model}} \times d_{\text{model}}}$. These projections are computed as:

$$\boldsymbol{Q}^l = \boldsymbol{Z}^{(l-1)} \boldsymbol{W}^{(Q,l)}, \quad \boldsymbol{K}^l = \boldsymbol{Z}^{(l-1)} \boldsymbol{W}^{(K,l)}, \quad \boldsymbol{V}^l = \boldsymbol{Z}^{(l-1)} \boldsymbol{W}^{(V,l)}.$$

The self-attention scores at layer $l$ are computed as the matrix $\boldsymbol{S}^{(l)} \in \mathbb{R}^{T \times T}$:

$$\boldsymbol{S}^{(l)} = \text{softmax}\left( \frac{\boldsymbol{Q}^l (\boldsymbol{K}^l)^\top}{\sqrt{d_{\text{model}}}} + \boldsymbol{M} \right), \tag{2}$$

where $\boldsymbol{M} \in \mathbb{R}^{T \times T}$ is an optional masking matrix, and the softmax is applied row-wise. The latent representation at layer $l$ is updated through the residual attention mechanism, defined as:

$$\boldsymbol{Z}^{(l)} = \tilde{\boldsymbol{Z}}^{(l)} + \boldsymbol{Z}^{(l-1)}, \quad \text{where} \quad \tilde{\boldsymbol{Z}}^{(l)} = \boldsymbol{S}^{(l)} \boldsymbol{Z}^{(l-1)} \boldsymbol{W}^{(V,l)}. \tag{3}$$

If skip connections are omitted, the update simplifies to $\boldsymbol{Z}^{(l)} = \tilde{\boldsymbol{Z}}^{(l)}$. By unrolling Eq. (3), we can derive formulations that relate the final latent representation, to useful statistical models. In the absence of skip connections, the latent representation at time step $t$ can be written as:

$$\boldsymbol{Z}_t = \boldsymbol{A}_t \tilde{\mathbf{Y}}_{[t]} \boldsymbol{B}, \tag{4}$$

where $\boldsymbol{A}_t = \boldsymbol{S}_{t,:}^{(L)} \boldsymbol{S}^{(L-1)} \cdots \boldsymbol{S}^{(1)} \in \mathbb{R}^{1 \times T}$, $\boldsymbol{B} = \boldsymbol{W}^{(V,1)} \cdots \boldsymbol{W}^{(V,L)} \in \mathbb{R}^{d_{\text{model}} \times d_{\text{model}}}$ and $\boldsymbol{S}_{t,:}^{(L)} \in \mathbb{R}^{1 \times T}$ denotes the $t$-th row of the final self-attention (SA) matrix $\boldsymbol{S}^{(L)}$. When skip connections are included (the standard case), the final representation becomes

$$\boldsymbol{Z}^{(L)} = \tilde{\mathbf{Y}}_{[t]} + \sum_{\emptyset \neq I \subseteq \{1,\ldots,L\}} \left( \prod_{i \in I^\downarrow} \boldsymbol{S}^{(i)} \right) \tilde{\mathbf{Y}}_{[t]} \left( \prod_{i \in I^\uparrow} \boldsymbol{W}^{(V,i)} \right), \tag{5}$$

where $I^\downarrow$ and $I^\uparrow$ denote the indices in descending and ascending order, respectively. Based on Eqs (4) and (5), the transformation of the input data in the Transformer is expressed as a left multiplication of the self-attention matrices across layers and a right multiplication of the Value projection matrices. The former is input-dependent, while the latter are learned parameters independent of the data. These two equations form the basis for the following proposition, which defines the structure of the self-attention latent space. An analytical proof is provided in Appendix G, along with additional details on the derived statements.

**Proposition 1** (Space-Time Autoregressive (STAR) structure of the self-attention latent space)**.**

1. *Without skip connections: Each time series in the Transformer's latent space follows a STAR-like structure. In particular, it can be expressed as $z_t^{(j)} = \sum_{k=1}^{d_{model}} b_{kj} \left( \sum_{q=1}^{t} a_{tq} \tilde{y}_q^{(k)} \right)$, which has the exact same form as the classical STAR model (Cressie & Wikle, 2011). The key distinction is that, while traditional STAR models use fixed lag weights $a_{tq}$ estimated by minimizing a loss function (e.g., mean squared error), the Transformer computes these weights dynamically from the input through the attention mechanism, with Q and K guiding their estimation in real time.*

2. **With skip connections:** *Each time series in the final representation, Eq. (5), can be interpreted as a **linear combination of multiple STAR-like processes**, where each component captures distinct temporal and feature-level dependencies of the original MTS.*

3. **With feed-forward layers:** *Adding feed-forward layers does not alter the STAR-like structure of the latent space. The only difference is that the spatial weights take a more complex form, as explained analytically in Proof G.*

**Linear Projection for the Reconstruction:** The final reconstructed MTS at time step $t$ is computed by applying a linear projection to the latent representation at the same time step:

$$\hat{\boldsymbol{Y}}_t = \boldsymbol{Z}_t^{(L)} \boldsymbol{W}^{\text{out}} \quad \Longrightarrow \quad \hat{y}_t^{(k)} = \sum_{j=1}^{d_{\text{model}}} w_{jk}^{\text{out}} z_t^{(L,j)}, \tag{6}$$

where $\boldsymbol{W}^{\text{out}} \in \mathbb{R}^{d_{\text{model}} \times d}$ is the output projection matrix that maps the final latent representation to the original input space. This reveals that the model learns to reconstruct each time series using a linear combination of multiple STAR processes (Proposition 1).

## 5 PROPOSED ANOMALY DIAGNOSIS METHOD

This section presents the proposed anomaly diagnosis method. Section 5.1 introduces the ALoRa-T model and its associated anomaly detection score, which together form the ALoRa-Det module, with an overview provided in Fig. 2. Section 5.2 then presents the ALoRa-Loc method for anomaly localization, illustrated in Fig. 1. The corresponding pseudocodes are provided in Appendix F and a computational analysis of the model is provided in Appendix C.

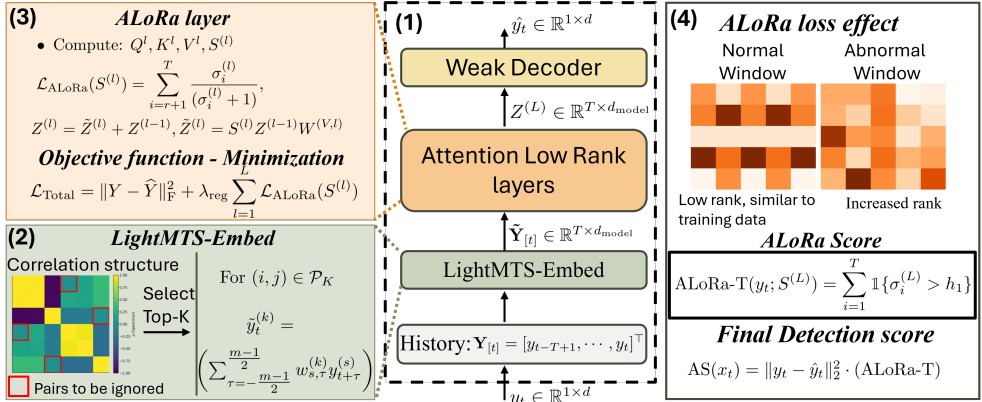

Figure 2: Overview of ALoRa-T and ALoRa-Det. (1) The architecture comprises the LightSMTS-Embed module, ALoRa layers, and a decoder. (2) The embedding module exploits correlation structures to retain only the most significant time series pairs, significantly reducing complexity by avoiding unnecessary information. (3) ALoRa layers impose a low-rank constraint on the self-attention matrix through a novel loss and a regularization term in the objective function, producing a signal for abnormality. (4) During inference, anomalous windows yield higher attention ranks, which are captured by the ALoRa score, providing a clear indicator of anomalies.

### 5.1 ATTENTION LOW-RANK TRANSFORMER FOR ANOMALY DETECTION (ALORA-DET)

ALoRa-Det consists of the ALoRa-T architecture together with its detection scoring method. We begin by describing the ALoRa-T architecture. The model begins with a lightweight MTS embedding module, followed by multi-head low-rank self-attention layers (ALoRa layers) with skip connections. As stated in Proposition 1 (see the proof in Section G), feedforward layers do not alter the structure of the latent space and are therefore omitted to avoid unnecessary complexity. The reconstruction step is implemented through a linear projection layer.

**LightMTS-Embed:** As shown in Section 4, embedding with 1D convolutions is equivalent to applying VMA filtering to MTS. However, using fully dense filters that mix all input series across embedding dimensions is both computationally expensive and less interpretable. In practice, not all time series are correlated with one another, and ignoring this fact can obscure the underlying dynamics in subsequent layers while further reducing interpretability. To address this, **each convolutional kernel is restricted to aggregate information from exactly two input series**. Specifically, for each output time series $k$, the (sparse) kernel contains only two non-zero weights $w_{i,j}^{(k)}$, corresponding to a particular pair of input series. To avoid including pairs without strong dependence, only the top-$K$ pairs are retained, ranked by Spearman correlation on the training data (commonly $K = 512$). When the total number of possible pairs does not exceed $K$, all $\binom{d}{2}$ pairs are included. This design improves efficiency, promotes sparsity, and enhances interpretability while preserving performance. An ablation study on the parameter efficiency and performance of the proposed embedding module, compared with the standard Transformer embedding, is provided in Appendix D.1.

**Low-Rank Regularization in Self-Attention:** Based on Eq. (4), since the SA-matrices are the only input-dependent learnable components, their spectral properties can serve as informative signals for anomaly detection. In particular, the *rank* of a matrix, defined as the number of non-zero singular values, provides a useful indicator of abnormal behavior. Empirically, we observe that the rank of SA-matrices increases in the presence of anomalies, as illustrated in Fig. 3 (right). Motivated by this observation, we propose **ALoRa layers**, which extend the standard self-attention mechanism by explicitly promoting a low-rank structure in the attention matrices. This is achieved through the *ALoRa loss*, a regularization term applied to each attention matrix $\boldsymbol{S}^{(l)}$, defined as the truncated Geman nuclear norm (Geman & Yang, 1995), which enforces singular values close to zero.

$$\mathcal{L}_{\text{ALoRa}}(\boldsymbol{S}^{(l)}) = \sum_{i=r+1}^{T} \frac{\sigma_i^{(l)}}{(\sigma_i^{(l)} + 1)}, \tag{7}$$

where $\sigma_1^{(l)} \geq \cdots \geq \sigma_T^{(l)} \geq 0$ are the singular values of $\boldsymbol{S}^{(l)}$. The parameter $r$ specifies the number of leading singular values that are preserved without penalty. Since $\boldsymbol{S}^{(l)}$ is row-stochastic (Eq. 2) with fixed largest singular value $\sigma_1 = 1$, we set $r = 1$, since penalizing it is unnecessary. The ALoRa layer is extended to the multi-head attention (MHA) setting by defining $\boldsymbol{S}^{(l)} = \frac{1}{H} \sum_{h=1}^{H} \boldsymbol{S}_h^{(l)}$, where $\boldsymbol{S}_h^{(l)}$ denotes the self-attention matrix of head $h = 1, \ldots, H$, and $H$ is the number of heads.

**Training:** The objective function, $\mathcal{L}_{\text{Total}}$, used to train the model includes two key components:

$$\mathcal{L}_{\text{Total}} = \|\boldsymbol{Y} - \widehat{\boldsymbol{Y}}\|_{\text{F}}^2 + \lambda_{reg} \sum_{l=1}^{l=L} \cdot \mathcal{L}_{\text{ALoRa}}(\boldsymbol{S}^{(l)}) \quad . \tag{8}$$

The first term penalizes the Frobenius norm of the reconstruction error, while the second term encourages a low-rank structure by summing the ALoRa losses of the SA-matrices $\boldsymbol{S}^{(l)}$ across all layers. The regularization strength is controlled by the parameter $\lambda_{\text{reg}}$. The effectiveness of the proposed low-rank regularization is illustrated in Fig. 3, which compares the rank of self-attention matrices with and without regularization. In both cases, rank differences between normal and anomalous inputs are evident but become substantially more pronounced with regularization. Consequently, anomalous patterns are more easily distinguished, providing a new abnormality signal.

**Model inference - Anomaly detection:** As in training, the MTS is processed in overlapping windows to ensure fair reconstruction across all time steps, enabling unbiased anomaly detection (Toscano & Recchioni, 2022). The detection anomaly score (AS) for the time steps $t$ is defined as:

$$\text{AS}(y_t) = \|y_t - \hat{y}_t\|_2^2 \cdot (\text{ALoRa-T}) \in \mathbb{R}, \tag{9}$$

$$\text{ALoRa-T}(y_t; \boldsymbol{S}^{(L)}) = \sum_{i=1}^{T} \mathbf{1}_{\{\sigma_i^{(L)} > h_1\}} \in \mathbb{R}, \tag{10}$$

where $\{\sigma_1^{(L)}, \ldots, \sigma_k^{(L)}\}$ are the singular values of the SA-matrix $\boldsymbol{S}^{(L)}$, used during the encoding-decoding of the MTS at time step $t$. The indicator function $\mathbf{1}_{\{\cdot\}}$ returns 1 if the condition is true and 0 otherwise. The threshold $h_1$ is introduced because, while the low-rank regularization encourages many singular values to be close to zero, they are often not exactly zero. Moreover, when the

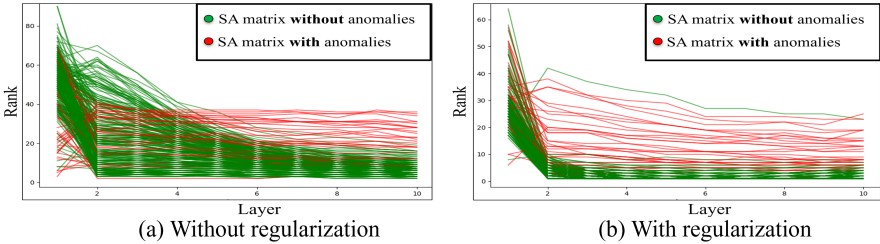

(a) Without regularization        (b) With regularization

Figure 3: Layer-wise rank of SA-matrices across transformer layers, without (left) and with (right) the low-rank regularization term, on the SMD dataset. The discrepancy in rank between normal and anomalous inputs becomes more pronounced with regularization, improving anomaly sensitivity.

anomaly score exceeds a threshold, $AS(y_t) > h_2$, an anomaly alarm is triggered. Details about $h_1$ and $h_2$ selection are provided in Appendix A. The ALoRa-T score captures the temporal characteristics of anomalies, providing earlier and more reliable indications than other scoring functions, as shown in Figure 4 (see also Figs. 12 and 11 in Appendix F).

## 5.2 ALoRa-Loc for MTS anomaly localization and interpretations

Effective anomaly localization in MTS requires a deep understanding of the model's learning process. Practitioners often ask: *How does the model make decisions, and which time series influence each output?* However, due to a limited understanding of the learning dynamics, such questions remain largely unanswered in the context of deep learning. In contrast, linear regression remains popular for its simplicity and interpretability. Each reconstructed value follows $\widehat{x_i} = \sum_j c_{ij} x_j + b_i$, where $c_{ij}$ quantifies the influence of input $x_j$ on output $\widehat{x_i}$.

The proposed localization method (ALoRa-Loc) first addresses this gap by deriving weights that capture how each input time series contributes to both the learned latent representation and the reconstruction of each output time series. This is particularly important, since during model inference one can quantify how strongly the model relies on a given time series $i$, as measured by its relative contribution with magnitude $C_{ij}/\sum_k C_{ik}, j \in \{1, \cdots, d\}$. For the anomaly localization task, such interpretability of the model's decision process is even more critical. As demonstrated in Appendix E, anomalies in one series can propagate to others during the modeling process, making it essential to capture these propagation effects. The contribution weights $E_{ij}$ and $C_{ij}$ provide exactly this capability: $E_{ij}$ traces the influence of input series $i$ on the latent features, while $C_{ij}$ extends this influence to the reconstructed outputs. By leveraging these weights, ALoRa-Loc enables tracing anomalies backward from the reconstructions to the latent space and ultimately to the originating input series, thereby supporting accurate anomaly localization. The derivation of the contribution weights is based on the theoretical analysis in Section 4, with analytical details in Appendix G.1.

**Contribution of each input time series to the latent space:** From Eq. (4) and Eq. (5), the latent representation is governed by the product of two matrices. The left product captures the covariance contributions, whereas the right product corresponds to shared weights across all time series and therefore does not affect the variability of individual contributions (see Appendix G.1 for details).

Accounting for the embedding module, the final contribution weights from the input space to the latent space are given by:

$$E_{ij} = \sum_{k=1}^{d_{\text{model}}} \left( \sum_{l=-\frac{m-1}{2}}^{\frac{m-1}{2}} w_{i,l}^{(k)} \right) b_{kj}, \quad \boldsymbol{B} = \begin{cases} \prod_{i=1}^{L} \left( \boldsymbol{W}^{(V,i)} + I \right), & \text{with skip connections (Eq. (5))}, \\ \prod_{i=1}^{L} \boldsymbol{W}^{(V,i)}, & \text{without skip connections (Eq. (4))}. \end{cases} \tag{11}$$

Here, $b_{ij}$ denotes the $(i,j)$-th entry of $\boldsymbol{B} \in \mathbb{R}^{d_{\text{model}} \times d_{\text{model}}}$, and $w_{i,l}^{(k)}$ are the embedding filter weights for input series $i$ at lag $l$. In the absence of embedding, the weights reduce to $\boldsymbol{E} = \boldsymbol{B}$.

**Contribution of each input time series to the reconstruction space:** From Eq. (6), it follows that the overall contribution of the $i$-th input time series to the $j$-th reconstructed time series is given by:

$$C_{ij} = \sum_{k=1}^{d_{\text{model}}} w_{kj}^{\text{out}} \cdot E_{ik} \tag{12}$$

Based on these insights, we define the localization anomaly score (LAS) for each time series $i$ at time $t$ as:

$$LAS_t^{(i)} = \sum_{j=1}^{d} C_{ij} \|y_t^{(j)} - \hat{y}_t^{(j)}\|_2^2 \tag{13}$$

The intuition behind this score is that each term $(C_{ij}\|y_t^{(j)} - \hat{y}_t^{(j)}\|_2^2)$ represents the magnitude of the anomaly in the $i$-th time series that has propagated to the reconstruction of the $j$-th one. Summing over all $j$ captures the total influence of the anomaly in time series $i$ across the system. In practice, it is often more effective to sum only over the top-$k$ dimensions with the largest $C_{ij}$, focusing on the most influential components. We refer to this variant as ALoRa-Loc (top-$k$).

## 6 EXPERIMENTS

### 6.1 DATASETS, BASELINES AND EVALUATION METRICS

**Datasets & baselines:** We evaluated ALoRa on six widely used datasets from diverse domains: SWaT (Mathur & Tippenhauer, 2016) and HAI (Shin et al., 2021) from industrial control, SMD (Su et al., 2019) and PSM (Abdulaal et al., 2021) from IT monitoring, and MSL (Hundman et al., 2018) from spacecraft telemetry. For localization evaluation, we use SMD, SWaT, and the MSDS dataset (Nedelkoski et al., 2020). We compare ALoRa against a wide range of baselines, including classical methods (PCA (Shyu et al., 2003), KNN (Ramaswamy et al., 2000), IForest (Liu et al., 2008), LOF (Breunig et al., 2000), OC-SVM (Schölkopf et al., 2001)), notable deep-learning models (Omni-Anomaly (Su et al., 2019), Interfusion (Li et al., 2021), DAEMON (Chen et al., 2021), contrastive learning-based methods (Yang et al., 2023), and recent SOTA approaches such as Anomaly Transformer (A.T) (Xu et al., 2022), MEMTO (Song et al., 2023), NPSR (Lai et al., 2023), $D^3R$ (Wang et al., 2023), and SARAD (Dai et al., 2024). For localization we also compared with AERCA (Han et al., 2025). All results are from our own runs using official or public code with recommended settings. See Appendix B for details.

**Detection metrics:** Recent studies on MTS anomaly detection evaluation have shown that range-based metrics are the most appropriate, as they address the limitations of point-adjusted and point-wise evaluation methods (Liu & Paparrizos, 2024). Moreover, since MTS anomaly detection involves highly imbalanced datasets, $F_1$-score–based metrics are considered the most reliable. Accordingly, our evaluation focuses on the affiliation-based $F_1$-score, precision, and recall (Huet et al., 2022). For fair comparison, we report the best $F_1$-scores and the corresponding precision and recall, avoiding method-specific thresholds. Additional results using the range-based $F_1$-score $(RF_1)$ (Hwang et al., 2019), VUS-AUC (VA), and VUS-PR (VPR) (Paparrizos et al., 2022) are provided in Appendix F. Additional details on the selection of evaluation metrics are provided in Appendix B.3.

**Localization metrics:** We evaluate localization performance using standard metrics such as Hit Rate (Su et al., 2019) and Normalized Discounted Cumulative Gain (NDCG) (Järvelin & Kekäläinen, 2002). Additionally, we use the Interpretation Score (IPS), which measures how accurately anomalies are localized at the segment level. Further details are provided in Appendix B.3.

### 6.2 RESULTS

All experiments were conducted over five runs, and we report the average performance values. Due to space restrictions, the standard deviations are presented in Appendix F.

**Detection Results:** As shown in Table 1, ALoRa-Det outperforms all baselines on four out of five datasets and ranks second on SWaT, where it still significantly outperforms most other methods.

Compared to the second-best performing models, ALoRa-Det achieves absolute improvements in affiliation-based $F_1$-score of 11.5% on SMD, 7.8% on PSM, 5.9% on MSL, and 8.9% on the HAI dataset. These results highlight the effectiveness and generalizability of our detection approach. Notably, our method is not only highly accurate, but its ALoRa-T score also provides a highly informative anomaly signal (see Figure 4 and Appendix F) that captures the characteristics of anomalies and enables much earlier detection than competing methods in most cases.

Table 1: **Detection performance**. P, R and $F_1$ denote Precision, Recall, and $F_1$-score respectively. The best $F_1$-scores are highlighted in bold, and the second-best are underlined.

| Method | SMD | | | PSM | | | MSL | | | SWaT | | | HAI | | |
|---|---|---|---|---|---|---|---|---|---|---|---|---|---|---|---|
| | P | R | $F_1$ | P | R | $F_1$ | P | R | $F_1$ | P | R | $F_1$ | P | R | $F_1$ |
| KNN | 0.70 | 0.34 | 0.46 | 0.53 | 0.98 | 0.68 | 0.50 | 0.25 | 0.33 | 0.45 | 0.41 | 0.43 | 0.48 | 0.36 | 0.41 |
| PCA | 0.84 | 0.40 | 0.54 | 0.92 | 0.38 | 0.54 | 0.55 | 0.32 | 0.41 | 0.49 | 0.43 | 0.46 | 0.51 | 0.40 | 0.45 |
| LOF | 0.56 | 0.35 | 0.43 | 0.60 | 0.41 | 0.49 | 0.52 | 0.30 | 0.38 | 0.50 | 0.38 | 0.43 | 0.47 | 0.34 | 0.39 |
| OC-SVM | 0.44 | 0.28 | 0.34 | 0.88 | 0.47 | 0.62 | 0.53 | 0.29 | 0.38 | 0.46 | 0.36 | 0.40 | 0.42 | 0.30 | 0.35 |
| IsolationForest | 1.00 | 0.09 | 0.17 | 1.00 | 0.03 | 0.06 | 0.62 | 0.08 | 0.14 | 0.50 | 0.11 | 0.18 | 0.68 | 0.07 | 0.13 |
| OmniAnomaly | 0.68 | 0.66 | 0.67 | 0.70 | 0.64 | 0.67 | 0.5 | 0.93 | 0.64 | 0.49 | 0.52 | 0.50 | 0.51 | 0.39 | 0.44 |
| InterFusion | 0.67 | 0.66 | 0.67 | 0.69 | 0.65 | 0.67 | 0.51 | 0.92 | 0.64 | 0.47 | 0.40 | 0.43 | 0.50 | 0.93 | 0.64 |
| A.T. | 0.58 | 0.88 | 0.70 | 0.55 | 0.83 | 0.66 | 0.51 | 0.96 | 0.67 | 0.57 | 0.37 | 0.45 | 0.61 | 0.52 | 0.56 |
| DAEMON | 0.70 | 0.69 | 0.70 | 0.72 | 0.68 | 0.70 | 0.52 | 0.93 | 0.65 | 0.60 | 0.59 | 0.60 | 0.62 | 0.84 | 0.71 |
| DCdetector | 0.54 | 0.94 | 0.69 | 0.53 | 0.82 | 0.67 | 0.51 | 0.9 | 0.65 | 0.58 | 0.76 | 0.66 | 0.72 | 0.86 | 0.78 |
| MEMTO | 0.78 | 0.86 | 0.79 | 0.63 | 0.75 | 0.68 | 0.53 | 0.95 | 0.67 | 0.60 | 0.61 | 0.60 | 0.60 | 0.66 | 0.64 |
| NPSR | 0.77 | 0.98 | 0.87 | 0.67 | 0.89 | 0.76 | 0.52 | 0.98 | 0.68 | 0.98 | 0.16 | 0.28 | 0.98 | 0.65 | 0.79 |
| $D^3R$ | 0.77 | 0.99 | 0.87 | 0.63 | 0.96 | 0.76 | 0.65 | 0.63 | 0.64 | 0.65 | 0.77 | **0.71** | 0.74 | 0.87 | 0.79 |
| SARAD | 0.88 | 0.67 | 0.78 | 0.73 | 0.505 | 0.56 | 0.56 | 0.83 | 0.67 | 0.54 | 0.33 | 0.41 | 0.51 | 0.70 | 0.66 |
| **ALoRa-Det** | 0.97 | 0.98 | **0.97** | 0.82 | 0.83 | **0.82** | 0.57 | 0.98 | **0.72** | 0.68 | 0.67 | 0.68 | 0.98 | 0.76 | **0.86** |

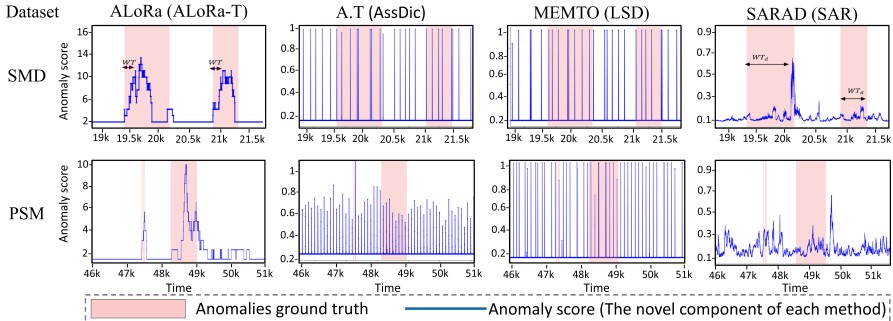

Figure 4: Anomaly scores for the SMD and PSM datasets. The red segment indicates the ground truth of anomalies. MEMTO and Anomaly Transformer behave close to random guessing, making detection unreliable. While SARAD provides more informative scores, it does not fully capture anomaly temporal patterns, often resulting in increased waiting time (WT) until detection. In contrast, ALoRa-T score captures these patterns effectively, enabling faster and more precise anomaly detection. Appendix F presents additional visualizations, including more datasets and methods.

**Computational Analysis:** To evaluate the practical efficiency of ALoRa-Det, we compare its model size (number of learnable parameters), training time, and inference time per sample with other SOTA methods. Due to page limitations, the detailed results are provided in Appendix C. As shown in Table 6, the proposed method is computationally efficient and suitable for real-world applications.

**Localization Results**: We use ALoRa-Loc (top-2) for SMD and the standard version for the other two datasets. Table 2 shows that ALoRa-Loc consistently outperforms the compared methods. Moreover, ALoRa-Loc provides meaningful interpretations of the learning process, which are essential for practitioners, as discussed in Section 5.2.

Table 2: **Localization performance**. We report HR@P%, NDCG@P%, and range-based IPS@P% for $P \in \{100, 150\}$. The best scores are shown in bold, with the second-best underlined. The Interfusion method applies only to segment-based localization, so only the IPS metric is reported.

| Method | SMD HR@P 100 | 150 | SMD NDCG@P 100 | 150 | SMD IPS@P 100 | 150 | MSDS HR@P 100 | 150 | MSDS NDCG@P 100 | 150 | MSDS IPS@P 100 | 150 | SWaT HR@P 100 | 150 | SWaT NDCG@P 100 | 150 | SWaT IPS@P 100 | 150 |
|---|---|---|---|---|---|---|---|---|---|---|---|---|---|---|---|---|---|---|
| MEMTO | 0.32 | 0.48 | 0.26 | 0.36 | 0.19 | 0.28 | 0.14 | 0.33 | 0.10 | 0.22 | 0.02 | 0.04 | 0.01 | 0.01 | 0.01 | 0.02 | 0.05 | 0.05 |
| OMNI | 0.29 | 0.46 | 0.24 | 0.34 | 0.17 | 0.26 | 0.13 | 0.35 | 0.09 | 0.21 | 0.01 | 0.03 | 0.01 | 0.01 | 0.01 | 0.01 | 0.04 | 0.04 |
| Interfusion | - | - | - | - | 0.59 | 0.75 | - | - | - | - | 0.03 | 0.05 | - | - | - | - | 0.10 | 0.12 |
| SARAD | 0.44 | 0.56 | 0.47 | 0.55 | **0.61** | 0.74 | 0.25 | 0.40 | 0.31 | 0.39 | 0.04 | 0.06 | 0.03 | 0.03 | 0.03 | 0.04 | 0.11 | 0.12 |
| DAEMON | 0.26 | 0.39 | 0.31 | 0.40 | 0.24 | 0.26 | 0.23 | 0.36 | 0.27 | 0.34 | 0.02 | 0.03 | 0.02 | 0.02 | 0.02 | 0.03 | 0.06 | 0.07 |
| AERCA | 0.21 | 0.25 | 0.36 | 0.45 | 0.13 | 0.17 | **0.33** | 0.56 | 0.31 | 0.40 | **0.08** | **0.12** | 0.013 | 0.014 | 0.031 | 0.04 | 0.028 | 0.029 |
| **ALoRa-Loc** | **0.56** | **0.76** | **0.60** | **0.70** | 0.60 | **0.81** | 0.30 | **0.57** | **0.32** | **0.45** | 0.03 | 0.05 | **0.042** | **0.068** | **0.041** | **0.056** | **0.16** | **0.20** |

# 7 ABLATION STUDIES

To verify the effectiveness of the individual components of the proposed method, we conduct a series of ablation studies. First, we evaluate the contribution of the proposed ALoRA loss, Eq. (7), and the corresponding ALoRA-T score, Eq. (10). The results of this analysis, presented in Table 3, demonstrate the effectiveness of both the ALoRA loss and the ALoRA-T score. Additional ablation studies exploring alternative implementations of the ALoRA loss are reported in Table 10 of Appendix D.3.

Table 3: **Ablation Study of ALoRA-Loss and ALoRA-Score Effectiveness:** We report the $F_1$ score for each case.

| Study Focus | ALoRA-Loss | ALoRA-Score | SMD | PSM | MSL | SWaT | HAI |
|---|---|---|---|---|---|---|---|
| **Effectiveness of ALoRA-Loss** | ✓ (Yes) | ✓ (Yes) | **0.97** | **0.82** | **0.72** | **0.68** | **0.86** |
| | ✗ (No) | ✓ (Yes) | 0.95 | 0.74 | 0.69 | 0.61 | 0.82 |
| **Effectiveness of ALoRA-Score** | ✓ (Yes) | ✗ (No) | 0.944 | 0.69 | 0.69 | 0.55 | 0.67 |

Next, for the Lightweight-MTS embedding module, we evaluate how specific design choices affect both performance and computational efficiency. In particular, we compare the model's performance when all time series pairs are used versus when only the Top-$K$ pairs are selected. The results presented in Table 4 illustrate that selecting only the top-$K$ pairs is more effective in terms of both performance and computational efficiency. Additional ablation studies for the Lightweight-MTS embedding module are provided in Appendix D.1. Finally, Appendix D.2 presents an ablation study supporting our decision to omit feed-forward layers. This choice is justified theoretically, since such layers do not alter the latent representation (Proposition 1), and is further confirmed experimentally, as shown in Table 9.

Table 4: **Ablation Study of LightMTS-Embed Design: Cost** : Computation Cost (Model Size in Millions (M) / Training Time (s) / Inference Time per Sample (ms)). **Det** : Anomaly detection performance measured using the $F_1$ score.

| Study Focus | Top-K | Correlation | SMD Cost | Det. | SWaT Cost | Det. | HAI Cost | Det. |
|---|---|---|---|---|---|---|---|---|
| **ALoRa-T** | ✓ (Top-K) | Spearman | 3.2M / 0.13 / 75 | 0.97 | 3.2M / 0.12 / 45 | 0.68 | 3.2M / 0.13 / 36 | 0.86 |
| **(A) All Pairs Instead of Top-K** | ✗ (All Pairs) | Spearman | 5.9M/ 0.19/ 90 | 0.96 | 19.5 / 0.25 / 62 | 0.68 | 108M/0.27/ 80 | 0.85 |
| **(B) Pearson Instead of Spearman** | ✓ (Top-K) | Pearson | 3.2M / 0.13 / 75 | 0.948 | 3.2M / 0.12 / 45 | 0.666 | 3.2M / 0.13 / 36 | 0.85 |

# 8 CONCLUSION

We study Transformer encoders on MTS and contribute theoretical insights into their learning behavior. Based on this, we propose ALoRa-Det for anomaly detection, and ALoRa-Loc for localization. Our methods achieve SOTA performance on both tasks. Future work includes extending our theoretical analysis from single-head to multi-head attention, and adapting our methods to settings where concept drift occurs alongside anomalous events.

## 9 REPRODUCIBILITY STATEMENT

To ensure the reproducibility of our results, the source code is available at: https://github.com/CharisShimillas/ALoRa , along with a README file that outlines the steps required to reproduce the experiments and results presented in this work. Appendix A provides details on parameter selection, while Appendix B describes the datasets used and the baseline methods. Datasets can be obtained from the sources listed in Appendix B.1. For each compared method, we provide source code links (Appendix B.2) and follow the recommended settings from the original studies.

## ACKNOWLEDGEMENTS

This work was supported by the European Union's Horizon Europe research and innovation programme under grant agreement No 101073307 (MSCA-DN LEMUR), the European Research Council (ERC) under grant agreement No 951424 (ERC-SyG Water-Futures), and the Republic of Cyprus through the Deputy Ministry of Research, Innovation and Digital Policy.

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

APPENDIX TO: LOW RANK TRANSFORMER FOR MULTIVARIATE TIME SERIES ANOMALY DETECTION AND LOCALIZATION

## A   TRAINING DETAILS

**ALoRa-T architecture design and hyperparameter selection:**    The ALoRa-T model consists of three self-attention layers with skip connections. Each layer uses $H = 8$ attention heads for all datasets. Training is performed using the ADAM optimizer with a learning rate of $10^{-4}$, and early stopping is applied to prevent overfitting. A fixed regularization parameter of $\lambda_{\text{reg}} = 10$ is used for all datasets.

**Window Size $T$:**    The window size $T$ is an important hyperparameter that determines how much past information the model can utilize at each time step. To assess its impact, we conduct an ablation study evaluating ALoRa-Det on the validation set with different values of $T$, as shown in Figure 5.

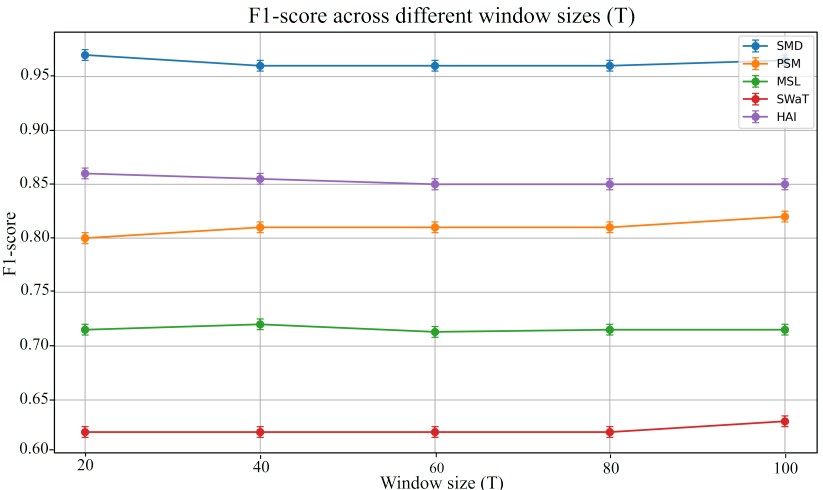

Figure 5: Validation F1-scores across different window sizes for each dataset. The average over 5 runs is reported along with the standard deviation.

The figure shows that ALoRa-Det's performance is not sensitive to the choice of window size. It is worth noting that smaller $T$ values lead to faster training and inference, while larger values result in longer processing times. Based on these considerations, we select the following window sizes for each dataset: SMD (20), PSM (100), MSL (20), SWaT (20), and HAI (20). For the MSDS dataset used for localization, the window size is set to 100.

**Selection of $h_1$, used in eq. (10):**    We present the process followed to select the threshold parameter $h_1$. As discussed in Section 5, ALoRa-T incorporates a low-rank regularization term, which encourages the self-attention matrices $S^{(L)}$ to have a reduced-rank structure. The selection procedure is automated and data-driven, as it must adapt to each dataset. With this procedure, $h_1$ is automatically determined for any new dataset based on the spectral properties of the self-attention matrices observed under normal operating conditions. Specifically, we analyze the distribution of the fourth and fifth largest eigenvalues of $S^{(L)}$ across the training sequence and track their trajectories. The threshold $h_1$ is then automatically chosen as the **maximum value** observed between these two eigenvalue trajectories. This ensures that only the dominant eigenvalues, typically corresponding to a rank of 3–4, are preserved. The choice to preserve the 3-4 largest eigenvalues is informed by the findings of Geshkovski et al. (2023). Figure 6 illustrates this selection process. Based on this analysis, we set $h_1 = 10^{-2}$ for SMD, $10^{-3}$ for PSM, $3 \times 10^{-2}$ for SWaT and MSL, and $2 \times 10^{-1}$ for HAI.

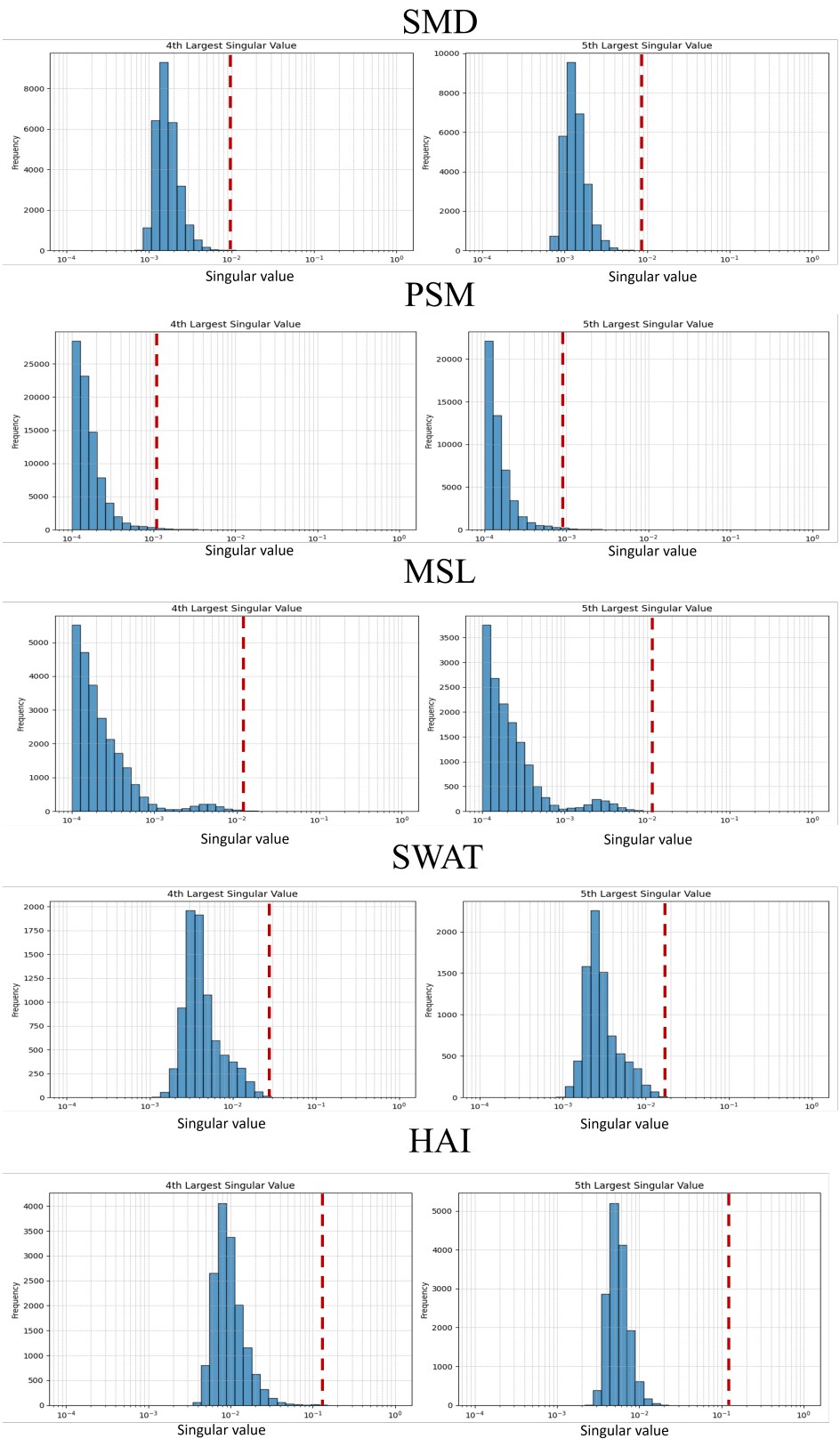

Figure 6: Distribution of the fourth and fifth largest eigenvalues of $S^{(L)}$ computed over normal training data. The red dashed line indicates the selected $h_1$ value, chosen to zero out smaller eigenvalues and ensure a typical rank no larger than 4.

**Selection of $h_2$:** The parameter $h_2$ is the threshold used to convert raw anomaly scores into binary decision labels. Such a threshold is required for **all** anomaly detection methods to provide anomaly predictions. To avoid introducing bias from any particular threshold-selection strategy and to ensure a **fair comparison across all methods**, we follow a widely adopted evaluation protocol: for each method, including our own, we report the performance obtained using the **best achievable threshold**, i.e., the threshold that yields the highest performance. This approach is standard in the anomaly detection literature (Dai et al., 2024; Lai et al., 2023). In summary, selecting $h_2$ in this manner guarantees **fairness** when comparing baselines: it evaluates the **quality of the anomaly scores produced by each method**, rather than the particular threshold-selection mechanism used to convert those scores into labels.

## B EXPERIMENTAL SETTING

### B.1 DATASETS

The datasets used in our experiments are publicly available and can be downloaded from the links provided below:

- **Secure Water Treatment (SWaT)** Li et al. (2019): This dataset captures data from a real-world industrial water treatment system monitored by 51 sensors over 11 days. The first 7 days contain normal operations, while the last 4 include 41 manually injected anomalies.
  *Download:* `https://itrust.sutd.edu.sg/itrust-labs-home/itrust-labs_swat/`

- **Server Machine Dataset (SMD)** Su et al. (2019): A large-scale dataset with 38 features collected over five weeks from server machines in a major Internet company, covering various system performance metrics.
  *Download:* `https://github.com/NetManAIOps/OmniAnomaly/tree/master/ServerMachineDataset`

- **Pooled Server Metrics (PSM)** Abdulaal et al. (2021): Collected internally at eBay, this dataset contains 25 dimensions from application server nodes, including 13 weeks of training data and 8 weeks of test data.
  *Download:* `https://github.com/eBay/RANSynCoders/tree/main/data`

- **Mars Science Laboratory (MSL)** Hundman et al. (2018): Provided by NASA, this dataset includes telemetry data with 55 features recorded from the MSL rover during space missions.
  *Download:* `https://github.com/khundman/telemanom`

- **HIL-based Augmented ICS Security Dataset (HAI)** Shin et al. (2021): Collected from a realistic ICS testbed enhanced with a Hardware-In-the-Loop simulator, this dataset contains 79 dimensions from sensors and actuators under normal conditions and during 38 simulated cyber-attacks.
  *Download:* `https://github.com/icsdataset/hai`

- **Multi-Source Distributed System (MSDS)** Nedelkoski et al. (2020): Collected from a cloud-based OpenStack testbed, this dataset contains multi-source monitoring data—including metrics, logs, and traces—under normal and fault-injected conditions. It includes root cause labels, making it suitable for anomaly localization task.
  *Download:* `https://zenodo.org/records/3484801`

Table 5 summarizes key statistics of the datasets used in our experiments, including the number of features, training and test samples, and the anomaly proportion in the test set.

Table 5: Summary of the five benchmark datasets used in our experiments. 'Dim' denotes the number of features. 'Train' and 'Test' indicate the number of samples in the training and test sets. 'Anomaly rate (%)' shows the percentage of anomalies in the test set.

|  | Dim | Application | Train | Test | *Anomaly rate(%)* |
|---|---|---|---|---|---|
| SWaT | 51 | Water | 495,000 | 449,919 | 0.121 |
| SMD | 38 | Server | 28479 | 28479 | 0.156 |
| PSM | 25 | Server | 132481 | 87841 | 0.278 |
| MSL | 55 | Space | 58317 | 73729 | 0.105 |
| HAI | 79 | Power | 921603 | 402005 | 0.223 |
| MSDS | 10 | AIOps | 29268 | 29286 | 0.72 |

**Dataset Preprocessing:** For each dataset, we first normalize each time series individually before passing the data to the model. This normalization is performed using the mean and standard deviation computed from the training set. The same normalization parameters are then applied to the

corresponding test set. For the SWaT and HAI datasets, which contain a very large number of time steps due to high-frequency (per-second) sampling, we downsample the data to one-minute intervals by averaging the values over every 60-second window.

## B.2 BASELINES

We reproduced the results of all baseline methods using their official or publicly available implementations on GitHub. For each baseline, we followed the configuration settings recommended in their respective papers to ensure optimal performance. The baselines and their implementations can be found at the following links:

- **kNN**: `https://github.com/yzhao062/pyod`
- **PCA**: `https://github.com/yzhao062/pyod`
- **LOF**: `https://github.com/yzhao062/pyod`
- **OCSVM**: `https://github.com/yzhao062/pyod`
- **IForest**: `https://github.com/yzhao062/pyod`
- **OmniAnomaly**: `https://github.com/NetManAIOps/OmniAnomaly`
- **InterFusion**: `https://github.com/ryu-ichiro/InterFusion`
- **Anomaly Transformer**: `https://github.com/thuml/Anomaly-Transformer`
- **DAEMON**: `https://github.com/Sherlock-C/DAEMON`
- **DCdetector**: `https://github.com/DAMO-DI-ML/KDD2023-DCdetector`
- **MEMTO**: `https://github.com/dreamgonfly/memto`
- **NPSR**: `https://github.com/lai-chihyu/NPSR`
- **SARAD**: `https://github.com/ZhihaoDai/SARAD`
- **D3R**: `https://github.com/Wang-Xinyu666/D3R`

## B.3 EVALUATION METRICS

**Detection:** The evaluation of multivariate time series anomaly detection has attracted growing research attention, largely due to the challenge of measuring model performance in a manner that aligns with real-world temporal decision-making. Anomalies usually span continuous ranges rather than isolated points. As a result, point-wise evaluation that treats each timestamp independently is inappropriate because it ignores temporal continuity and unfairly penalizes slight delays or near-misses in detection. Traditionally, the point-adjustment method was used to address this limitation, where detecting any point within an anomalous segment suffices to consider the entire segment as an observed anomaly (Xu et al., 2022; Song et al., 2023; Shen et al., 2020). However, this approach often inflates performance metrics, as even a random anomaly scoring method can achieve performance comparable to that of a well-informed method, which effectively captures the temporal characteristics of anomalies (Kim et al., 2022; Huet et al., 2022). This limitation of point-adjusted metrics is further illustrated in Figure 7.

Recognizing these limitations, many studies (Liu & Paparrizos, 2024; Hwang et al., 2019) have suggested range-based evaluation metrics as the most appropriate choice for multivariate time series anomaly detection. For instance, Hwang et al. (2019) proposed the range-based $F_1$, precision, and recall ($RF_1$, R-P, R-R). An advancement over these are the affiliation-based F1, precision, and recall (Huet et al., 2022), which demonstrate superior performance and robustness compared to the aforementioned metrics. Unlike traditional metrics, this approach accounts for the temporal alignment between predicted and actual anomalies. Precision is computed as the average directed distance from predicted anomaly events to their nearest ground truth counterparts, while recall measures the average directed distance from ground truth events to their closest predictions. For a more detailed explanation of how these metrics are computed, we refer the reader to the original paper (Huet et al., 2022). Paparrizos et al. (2022) introduced VUS-ROC and VUS-PR as range-aware alternatives to traditional AUC. However, because MTS anomaly detection is a highly imbalanced classification problem, $F_1$-like metrics are the most appropriate and reliable for evaluation, while VUS metrics can

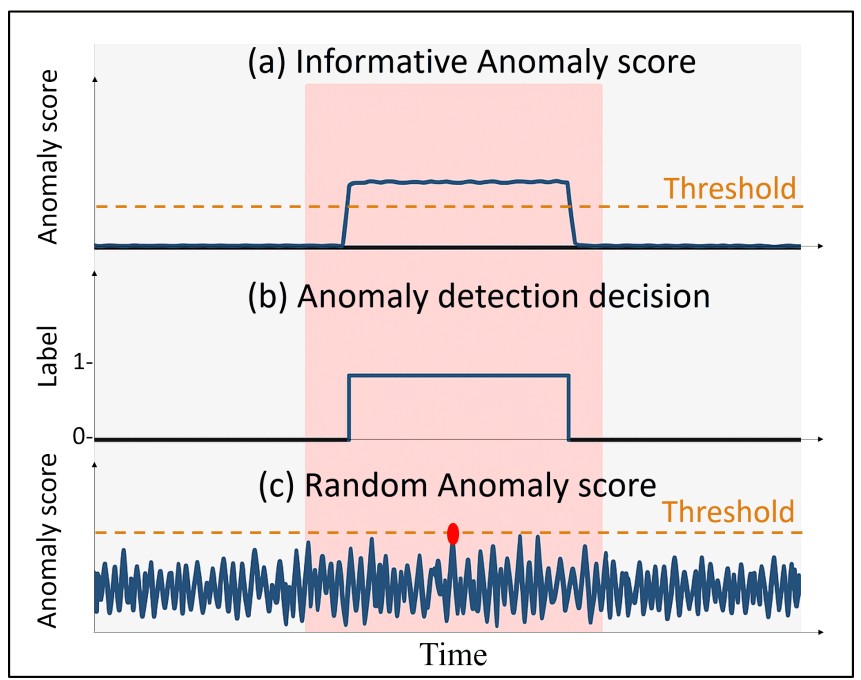

Figure 7: Line (a) shows the anomaly scores from an informative anomaly scoring approach, while Line (c) represents a random scoring method. The middle section displays the binary decisions (where anomaly = 1, otherwise 0) made by both methods using the point adjustment technique, which ultimately led to the same outcome. The red segments indicate the ground truth abnormal time steps, while the red dot in Line (c) marks the only time step that, by chance, exceeded the anomaly threshold.

provide additional insights but should be used together with F1-like metrics. Given the superior robustness and effectiveness of affiliation-based metrics, we adopt the affiliation-based $F_1$-score as our primary evaluation metric. Results for the range-based $F_1$-score ($RF_1$), along with the VUS-ROC ($V_{ROC}$) and VUS-PR ($V_{PR}$) metrics, are reported in Table 11. Nevertheless, due to the highly imbalanced nature of MTS anomaly localization, F1-like metrics remain the most appropriate choice, as they explicitly capture the trade-off between precision and recall.

**Localization:** At each time step $t$, anomaly localization is evaluated by identifying which features (time series) contribute to the detected anomaly. The ground-truth for localization is provided as a binary label vector $G_i \in \{0,1\}^N$, where $G_i = 1$ indicates that the $i$-th feature is anomalous at time $t$, and $G_i = 0$ otherwise.

After computing the anomaly score $AS_t^{(i)}$ for each feature using eq.( 13), features are ranked in descending order of their scores to identify those most likely responsible for the anomaly. The top-$k$ features are selected as:

$$\Gamma_{t@P\%} = \text{Top-}k \text{ features ranked by } AS_t^{(i)}, \quad \text{where } k = \lceil |G_i| \times P\% \rceil$$

For example, if there are 3 anomalous features (i.e., $|G_i| = 3$) and we evaluate at $P = 150$, then $k = 5$. The localization performance is then quantified using the *Hit Rate (HR)* at $P\%$, which measures the fraction of truly anomalous features that appear in the top-$k$ predictions:

$$\text{HR}_{t@P\%} = \frac{|G_i \cap \Gamma_{t@P\%}|}{|G_i|} \tag{14}$$

**Normalized Discounted Cumulative Gain (NDCG):** Let $r_j \in \{0, 1\}$ indicate whether the $j$-th ranked feature in $\Gamma_{t@P\%}$ is truly anomalous (i.e., belongs to $G_i$). The *Discounted Cumulative Gain (DCG)* and its ideal counterpart (IDCG) are defined as:

$$\text{DCG}_{t@P\%} = \sum_{j=1}^{k} \frac{r_j}{\log_2(j+1)}$$

$$\text{IDCG}_t = \sum_{j=1}^{|G_i|} \frac{1}{\log_2(j+1)}$$

The resulting *NDCG* is:

$$\text{NDCG}_{t@P\%} = \frac{\text{DCG}_{t@P\%}}{\text{IDCG}_t}, \tag{15}$$

NDCG ranges from 0 to 1, with higher values indicating better-ranked localization.

**Interpretation Score (IPS):** To evaluate anomaly localization at the segment level, we use the *Interpretation Score (IPS)* Li et al. (2021). For each anomaly segment $S_i$, we first compute a single score for each feature by taking the maximum anomaly score within the segment:

$$AS_{S_i}^{(j)} = \max_{t \in S_i} AS_t^{(j)} \tag{16}$$

We then identify the top-ranked features as the predicted anomalous dimensions $P_{S_i}$, and compare them against the ground-truth anomalous features $G_{S_i}$. The IPS measures the average proportion of correctly predicted features across all segments:

$$\text{IPS} = \frac{1}{N} \sum_{i=1}^{N} \frac{|G_{S_i} \cap P_{S_i}|}{|G_{S_i}|}, \tag{17}$$

where $N$ is the total number of anomalous segments. Each segment is equally weighted in the final score.

## C   COMPUTATIONAL ANALYSIS

To evaluate the practical efficiency of **ALoRa-Det**, we compare its model size (number of learnable parameters), total training time (in seconds), and inference time per sample (in milliseconds) against three state-of-the-art baselines: MEMTO, SARAD, and $D^3R$. The results across three benchmark datasets are reported in Table 6.

Table 6: Computational efficiency comparison. The table reports the number of learnable parameters (in millions), total training time (s), and inference time per sample (ms).

| Method | SMD (d=38) Params / Inf/ms / Train(s) | SWaT (d=51) Params / Inf/ms / Train(s) | HAI (d=79) Params / Inf/ms / Train(s) |
|---|---|---|---|
| MEMTO | 5.9M / 1.10 / 108 | 5.9M / 5.71 / 57 | 6M / 6.8 / 96 |
| SARAD | 9.6M / **0.11** / 147 | 9.6 / 0.16 / 652 | 9.6M / 0.48 / 126 |
| D3R | 52.3M / 4.83 / 994 | 52.3 / 6.27 / 283 | 52.5 M / 21.6 /1077 |
| ALoRa-Det | **3.2 M** / 0.13 / **75** | **3.2M** / **0.12** / **45** | **3.2M** / **0.13** / **36** |

The results in Table 6 indicate that **ALoRa-Det** almost always outperforms the competing methods across all three metrics. Unlike other approaches, the total number of parameters remains constant

as input dimensionality increases, while both inference time per sample and training time remain consistently low. These advantages become more evident as the dimensionality of the MTS grows, highlighting the scalability of the proposed method to high-dimensional settings. Taken together, these findings confirm that **ALoRa-Det** is not only more efficient than existing baselines but also highly suitable for real-time applications.

# D    ABLATION STUDIES - SUPPLEMENT

## D.1    LIGHTWEIGHT MTS EMBEDDING PARAMETER EFFICIENCY

Table 7 shows the parameter efficiency of the lightweight embedding used in our method, comparing it to standard fully dense filters.

Table 7: Parameter comparison between standard fully dense filters with model dimension, $d_{\text{model}} = 512$ and the proposed Lightweight MTS Embedding module.

| Input Size $d$ | Standard Weights | Custom Weights |
|---|---|---|
| 10 | 15,360 | 90 |
| 20 | 30,720 | 380 |
| 40 | 61,440 | 1,024 |
| 50 | 76,800 | 1,024 |
| 100 | 153,600 | 1,024 |
| 200 | 307,200 | 1,024 |

In addition, we compare the standard and Lightweight-MTS embedding methods in terms of detection performance to demonstrate that the proposed lightweight approach does not compromise effectiveness. On the contrary, it maintains, and in some cases even improves performance. Figure 8 presents the $F_1$ scores achieved by both embeddings, along with boxplots showing the distribution of reconstruction errors on normal test samples, providing further insight into the reconstruction behavior of each method.

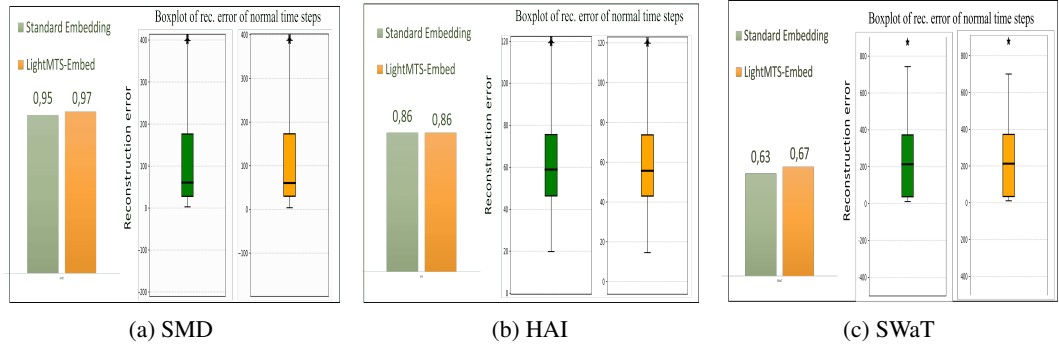

(a) SMD                          (b) HAI                          (c) SWaT

Figure 8: Model performance under the standard and lightweight embeddings for the SMD, HAI, and SWaT datasets. In each subplot, the figure on the left shows the $F_1$ score comparison, and the figure on the right presents the boxplot of reconstruction errors on normal test samples.

**Why the LightMTS embedding module use pairwise interactions:**    The proposed LightMTS embedding module was designed to reduce complexity by restricting interactions to **pairwise** relationships between time series. Increasing the number of time series involved in each interaction, from pairs to triplets or higher-order groups, would lead to a **combinatorial explosion** in the number of possible interactions. In such cases, the complexity would become comparable to that of standard fully connected or convolutional embeddings.

**Top-$K$ Selection for the LightMTS-Embed Module:** The choice to select the top-512 pairs follows standard design practice in Transformer architectures, where the embedding dimension is commonly set to 512, as introduced in the original Transformer model (Vaswani et al., 2017). This setting provides a strong balance between representational capacity and computational efficiency, enabling consistently low training and inference times across all datasets (Table 6). To assess the sensitivity of the model to this parameter, an ablation study for different values of $K$, is presented in Table 8. The results indicate that the model is not highly sensitive to this hyperparameter, as performance remains similar for $K = 512$ and $K = 256$, with the latter offering slightly lower accuracy but improved computational efficiency. However, as the model's computational efficiency is already very strong, as demonstrated in Table 6, $K = 512$ remains a robust default choice. It generalizes well across datasets from diverse domains and provides sufficient representation-learning capacity even for multivariate time-series datasets with very high dimensionality.

Table 8: **Ablation Study on the Top-$K$ pairs selection:** The table reports the $F_1$ score average over 5 runs with the corresponding standard deviations for different values of the parameter K.

| K | SMD | PSM | MSL | SWaT | HAI |
|---|---|---|---|---|---|
| **512** | $0.97 \pm 0.012$ | $0.82 \pm 0.014$ | $0.72 \pm 0.011$ | $0.68 \pm 0.010$ | $0.86 \pm 0.015$ |
| **256** | $0.965 \pm 0.014$ | $0.82 \pm 0.014$ | $0.72 \pm 0.011$ | $0.67 \pm 0.011$ | $0.854 \pm 0.015$ |
| **128** | $0.95 \pm 0.012$ | $0.805 \pm 0.015$ | $0.713 \pm 0.012$ | $0.66 \pm 0.012$ | $0.85 \pm 0.013$ |
| **64** | $0.94 \pm 0.014$ | $0.78 \pm 0.015$ | $0.67 \pm 0.012$ | $0.65 \pm 0.013$ | $0.84 \pm 0.012$ |

### D.2 ABLATION STUDY FOR REMOVING FEED-FORWARD LAYERS

This ablation study examines how omitting feed-forward layers affects both computational efficiency and anomaly detection and localization performance. As stated in Proposition 1, feed-forward layers do not alter the latent representation, indicating that their contribution to performance is minimal. The experimental results in Table 9 confirm this theoretical insight. Specifically, when feed-forward layers are included, performance either decreases or remains unchanged, while computational cost increases substantially.

Table 9: **Ablation Study for Removing Feed-Forward Layers**: The table reports the number of learnable parameters in millions (Cost), the anomaly detection performance using the $F_1$-score (Det.), and the localization performance using the Hit-Rate@100 metric (Loc.), reported only for the datasets where localization labels are available.

| | SMD | | | PSM | | MSL | | SWaT | | | HAI | |
|---|---|---|---|---|---|---|---|---|---|---|---|---|
| FFN | Cost | Det. | Loc. | Cost | Det. | Cost | Det. | Cost | Det. | Loc. | Cost | Det. |
| ✓ (Yes) | 6.3 | 0.96 | **0.56** | 2.2 | 0.79 | 6.3 | 0.72 | 6.3 | 0.67 | 0.03 | 6.3 | 0.854 |
| ✗ (No) | **3.2** | **0.97** | 0.50 | **1.1** | **0.82** | **3.2** | 0.72 | **3.2** | 0.68 | **0.042** | **3.2** | 0.86 |

### D.3 ALORA-LOSS PLACEMENT

The proposed ALoRa loss (Eq. 7) is applied to the average self-attention matrix across heads rather than to each head individually. To justify this choice, we provide an ablation study comparing the two approaches in terms of both computational cost and performance.

As shown in Table 10, both variants achieve nearly identical detection performance across all datasets. However, applying the ALoRa loss to the *average* self-attention matrix is slightly more computationally efficient, which is why it was preferred. Importantly, both approaches are capable of achieving our goal of reducing the rank while maintaining the effectiveness of the model. As a final note, even if the head-wise application of the ALoRa loss had been preferred, the proposed method would still remain more computationally efficient than other state-of-the-art methods.

Table 10: **Ablation Study on ALoRA-Loss Placement:** Head-Wise vs Averaged Self-Attention. Reported are the training time (in seconds) required for each case and the corresponding detection performance (F1-score).

| ALoRA-Loss Placement | SMD | | SWaT | | HAI | |
|---|---|---|---|---|---|---|
| | Training time | $F_1$ | Training time | $F_1$ | Training time | $F_1$ |
| Head-Wise ALoRA-Loss | 155 | 0.97 | 98 | 0.67 | 78 | 0.86 |
| OURS: Averaged Attention ALoRA-Loss | **75** | 0.97 | **0.45** | **0.68** | **0.36** | 0.86 |

## E    ANOMALOUS EFFECT ATTRIBUTION VIA EFFECT WEIGHTS

To examine how anomalies in one time series affect others through shared latent representations, we design a controlled simulation using a simple self-attention model. This setup also demonstrates how the *effect weights* introduced in Section 4 quantify influence in both latent and reconstruction spaces.

We generate bivariate i.i.d. data $X \in \mathbb{R}^{T \times 2}$, where:

$$x_t^{(1)} \sim \begin{cases} \mathcal{N}(\mu_1 + \Delta, \sigma_1^2), & \text{for } t_1 \leq t < t_2 \\ \mathcal{N}(\mu_1, \sigma_1^2), & \text{otherwise} \end{cases} , \quad x_t^{(2)} \sim \mathcal{N}(\mu_2, \sigma_2^2) \quad \forall t$$

We set $T = 500$, $t_1 = 200$, and $t_2 = 300$. The model consists of a 2-layer, 1-head self-attention mechanism. We fix the value matrices $W_V^{(l)}$ for layers $l = 1, 2$, as well as the output projection matrix $W_{\text{out}}$, while the remaining parameters are kept learnable:

$$W_V^{(1)} = I_2, \quad W_V^{(2)} = \begin{bmatrix} 0.2 & 0.7 \\ 0.8 & 0.3 \end{bmatrix}, \quad W_{\text{out}} = \begin{bmatrix} 0.1 & 0.9 \\ 0.9 & 0.1 \end{bmatrix}$$

This controlled setup allows us to observe how a localized anomaly in one input series propagates through the latent space and affects the reconstructed time series. As shown in Fig. 9, the effect weights $E_{1j}$ and $C_{1j}$ reflect how the anomaly in the first input series propagates through the latent space and influences the reconstructed time series, respectively. Together, they provide meaningful and interpretable attributions of anomaly influence across the model's internal and output representations.

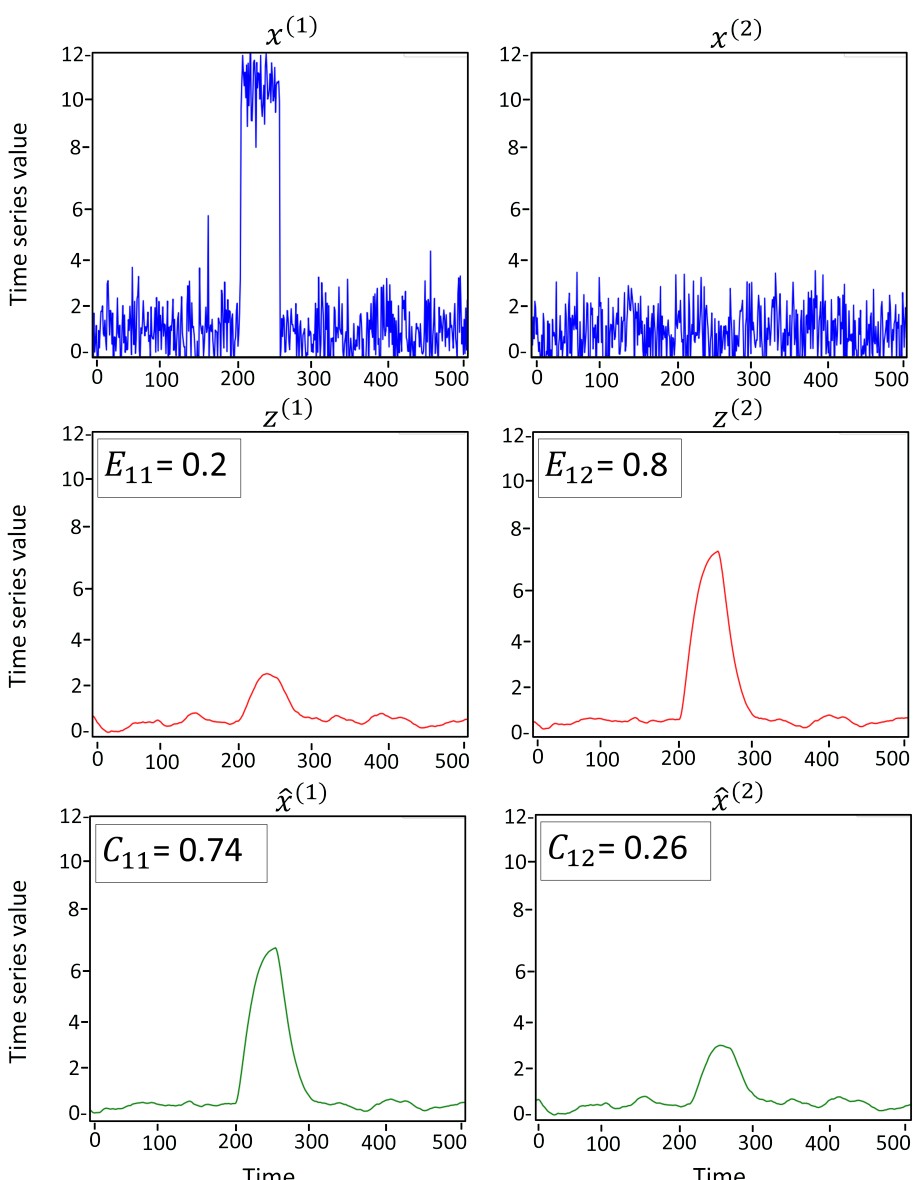

Figure 9: Averaged results over 100 simulation runs. **Top:** Input time series with an anomaly introduced in series 1 between time steps 200 and 300. **Middle:** Latent representations; effect weights $E_{1j}$ quantify the contribution of the anomaly in series 1 to each latent dimension. **Bottom:** Reconstructed time series; effect weights $C_{1j}$ reflect how the anomaly influences the reconstruction of each output series.

## F  EXPERIMENTS - SUPPLEMENT

To assess the performance of the proposed ALORA-DET method, we provide additional results using three evaluation metrics. The selection of these metrics is explained in Appendix B.3. Table 11 presents the performance of the proposed method and all compared methods using the range-based F1-score ($R_{F1}$), as well as the VUS-ROC ($V_{\mathrm{ROC}}$) and VUS-PR ($V_{\mathrm{PR}}$).

**Effectiveness of the ALoRA-T Score Across Anomaly Types:**  To assess the effectiveness of the ALoRA-T score, and to further provide empirical evidence that the attention rank increases in the presence of anomalies, we examine how the rank of the metric—and thus the behavior of

Table 11: **Range-aware evaluation metrics.** $RF_1$ denotes Range-based F1-score; $V_{\text{ROC}}$ and $V_{\text{PR}}$ denote the Volume Under the Surface ROC and precision, respectively. Best scores are highlighted in bold, and second-best are underlined.

| | SMD | | | PSM | | | MSL | | | SWaT | | | HAI | | |
|---|---|---|---|---|---|---|---|---|---|---|---|---|---|---|---|
| **Method** | $RF_1$ | $V_{\text{ROC}}$ | $V_{\text{PR}}$ | $RF_1$ | $V_{\text{ROC}}$ | $V_{\text{PR}}$ | $RF_1$ | $V_{\text{ROC}}$ | $V_{\text{PR}}$ | $RF_1$ | $V_{\text{ROC}}$ | $V_{\text{PR}}$ | $RF_1$ | $V_{\text{ROC}}$ | $V_{\text{PR}}$ |
| KNN | 0.15 | 0.42 | 0.09 | 0.31 | 0.40 | 0.28 | 0.11 | 0.37 | 0.08 | 0.10 | 0.44 | 0.35 | 0.12 | 0.09 | 0.12 |
| PCA | 0.18 | 0.65 | 0.11 | 0.48 | 0.58 | 0.42 | 0.15 | **0.62** | **0.20** | 0.13 | 0.68 | 0.51 | 0.21 | 0.63 | 0.13 |
| IsolationForest | 0.19 | 0.68 | 0.1 | 0.23 | 0.54 | 0.33 | 0.14 | 0.57 | 0.17 | 0.13 | 0.42 | 0.12 | 0.17 | 0.69 | 0.12 |
| OC-SVM | 0.11 | 0.58 | 0.08 | 0.19 | 0.53 | 0.37 | 0.13 | 0.59 | 0.18 | 0.03 | 0.62 | 0.52 | 0.23 | 0.64 | 0.14 |
| OmniAnomaly | 0.17 | 0.33 | 0.07 | 0.14 | 0.29 | 0.05 | 0.12 | 0.50 | 0.11 | 0.021 | 0.37 | 0.12 | 0.20 | 0.73 | 0.22 |
| InterFusion | 0.18 | 0.36 | 0.08 | 0.19 | 0.34 | 0.12 | 0.14 | 0.53 | 0.12 | 0.09 | 0.40 | 0.14 | 0.28 | 0.75 | 0.23 |
| A.T. | 0.11 | 0.50 | 0.10 | 0.19 | 0.26 | 0.37 | 0.14 | 0.51 | 0.12 | 0.13 | 0.61 | 0.22 | 0.09 | 0.59 | 0.21 |
| MEMTO | 0.26 | 0.52 | 0.11 | 0.29 | 0.50 | 0.29 | 0.13 | 0.50 | 0.12 | 0.14 | 0.68 | 0.39 | 0.16 | 0.60 | 0.22 |
| NPSR | 0.43 | 0.89 | 0.48 | **0.49** | 0.63 | 0.47 | 0.20 | 0.53 | 0.13 | **0.33** | **0.83** | **0.64** | 0.20 | 0.75 | 0.44 |
| $D^3R$ | 0.1 | 0.91 | 0.57 | 0.2 | 0.65 | 0.46 | 0.18 | 0.61 | 0.15 | 0.09 | 0.84 | 0.38 | 0.1 | 0.89 | 0.61 |
| SARAD | 0.34 | 0.80 | 0.16 | 0.25 | 0.62 | 0.42 | 0.12 | 0.53 | 0.13 | 0.171 | 0.59 | 0.33 | 0.07 | 0.70 | 0.21 |
| **ALoRa-Det** | **0.50** | **0.93** | **0.58** | 0.45 | **0.69** | **0.50** | 0.28 | 0.58 | 0.15 | 0.23 | 0.78 | 0.57 | **0.62** | **0.92** | **0.62** |

the ALoRA-T score—changes under different anomaly types. These anomaly types are the ones formalized and defined in the studies (Lai et al., 2021) and (Tsay et al., 2000).

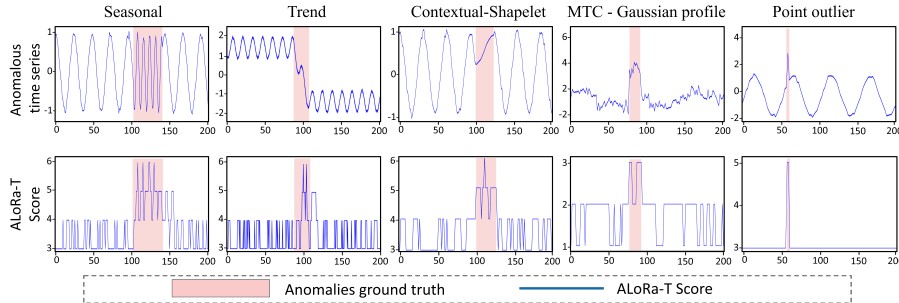

Figure 10: ALoRa-T scores for different anomaly types, following the definitions and categorizations presented in (Lai et al., 2021) and (Tsay et al., 2000). The plotted ALoRa-T scores are averaged over 5 runs.

Based on Fig. 10, there is clear evidence that, across a variety of anomaly types, the rank of the self-attention matrices, and consequently the corresponding ALoRA-T score, increases in the presence of anomalies. This consistent behavior across different anomaly categories demonstrates the robustness and generalizability of the ALoRA-T score in detecting anomalous patterns.

**Comparison of ALoRa-T Scores with State-of-the-Art Methods:** To support and extend the findings in the main paper, this appendix presents additional visualizations to demonstrate the generalizability of the ALoRa-T anomaly scoring method. First, we provide additional abnormal segments from the SMD and PSM datasets (Fig. 11) to reinforce our earlier results. Second, we include anomaly scores from the SWaT and HAI datasets (Fig. 12) to illustrate ALoRa-T's effectiveness across multiple domains. Based on Fig. 12 and 11, the ALoRa-T score continues to stand out for its effectiveness to capture the temporal characteristics of anomalies, providing timely and highly informative responses to anomalous patterns. The score rises quickly and remains aligned with the duration and structure of the anomalies. In some cases, the ALoRa-T score remains high for a short period of time, even after the labeled anomaly segment ends. However, this behavior can be meaningfully interpreted: the self-attention matrices at post-anomaly timesteps still incorporate abnormal information from earlier points. The score gradually decreases after $T$ timesteps—where $T$ is the window size, once the attention mechanism no longer includes the earlier abnormal data. This demonstrates the effectiveness of the ALoRa-T score to accurately detect anomalies and to provide interpretable signals for both true and false alarms. In contrast, MEMTO and Anomaly Transformer still behave similarly to their performance in the main results (see Fig. 4), where their outputs resemble random guessing. This makes their detection signals unreliable. SARAD and NPSR produce more informative patterns. However, they frequently fail to capture the temporal dynamics of anomalies accurately, often resulting in delayed or unsuccessful detection.

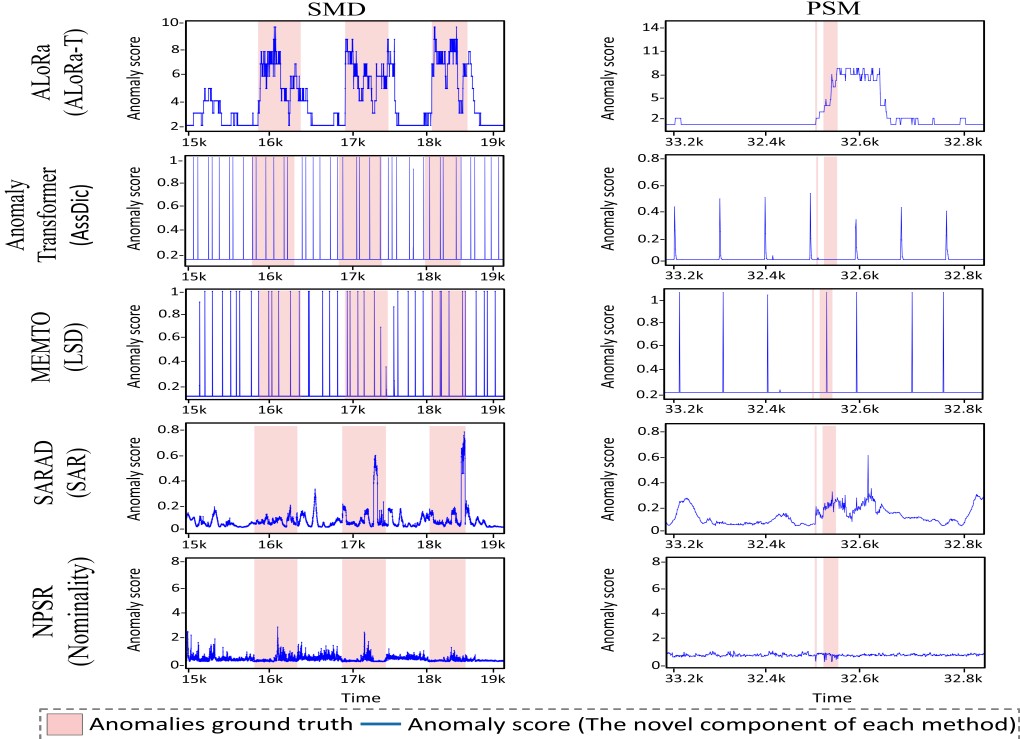

Figure 11: Anomaly scores for the SMD and PSM, datasets. The red segment indicates the ground truth of anomalies

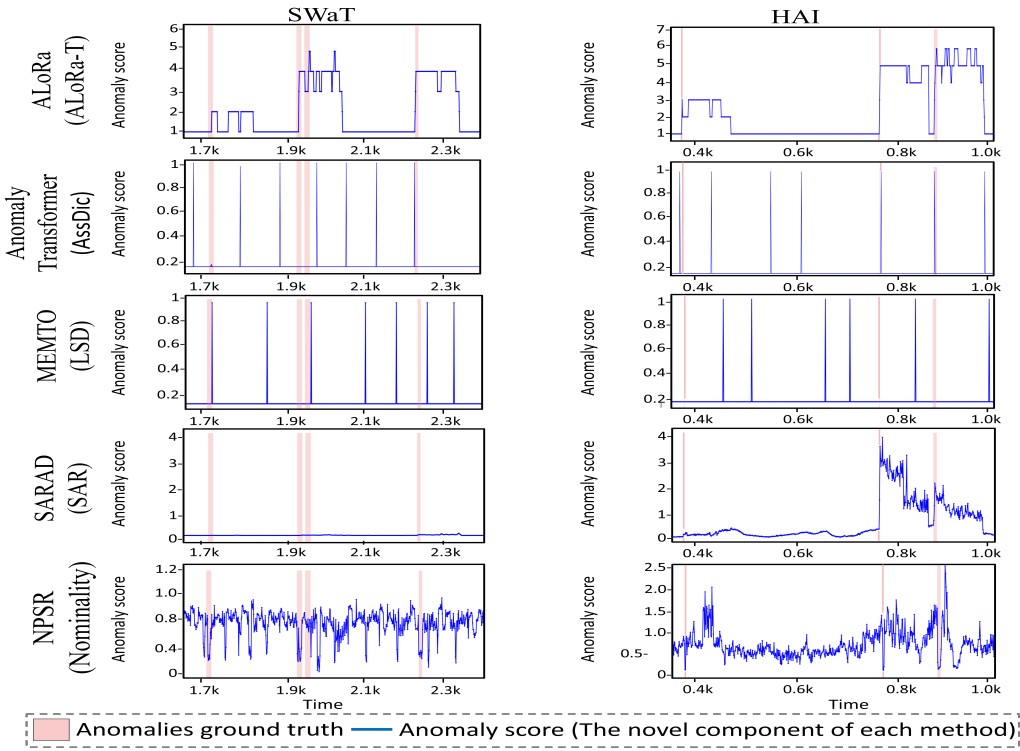

Figure 12: Anomaly scores for the SWat and HAI, datasets. The red segment indicates the ground truth of anomalies

**Standard Deviation of Anomaly Diagnosis Results:** We report the standard deviations of the anomaly diagnosis results presented in Section 6, as well as for the additional evaluation metrics presented in Table 11. Specifically, Table 12 shows the standard deviations of the detection results reported in Table 1, while Table 14 reports the standard deviations for the results in Table 11. Finally, Table 13 presents the standard deviations of the localization results shown in Table 2.

Table 12: **Standard deviation of detection metrics**, for the results in Table 1. P, R and $F_1$ denote the standard deviation of Precision, Recall, and $F_1$-score respectively.

| Method | SMD P | SMD R | SMD $F_1$ | PSM P | PSM R | PSM $F_1$ | MSL P | MSL R | MSL $F_1$ | SWaT P | SWaT R | SWaT $F_1$ | HAI P | HAI R | HAI $F_1$ |
|---|---|---|---|---|---|---|---|---|---|---|---|---|---|---|---|
| KNN | 0.006 | 0.004 | 0.005 | 0.003 | 0.002 | 0.002 | 0.009 | 0.006 | 0.008 | 0.003 | 0.002 | 0.002 | 0.006 | 0.004 | 0.005 |
| PCA | 0.009 | 0.007 | 0.008 | 0.006 | 0.004 | 0.005 | 0.003 | 0.002 | 0.002 | 0.009 | 0.007 | 0.008 | 0.003 | 0.002 | 0.002 |
| LOF | 0.003 | 0.002 | 0.002 | 0.009 | 0.007 | 0.008 | 0.006 | 0.004 | 0.005 | 0.006 | 0.005 | 0.005 | 0.003 | 0.002 | 0.002 |
| OC-SVM | 0.006 | 0.004 | 0.005 | 0.006 | 0.004 | 0.005 | 0.003 | 0.002 | 0.002 | 0.009 | 0.007 | 0.008 | 0.009 | 0.007 | 0.008 |
| IsolationForest | 0.009 | 0.007 | 0.008 | 0.003 | 0.002 | 0.002 | 0.006 | 0.004 | 0.005 | 0.006 | 0.004 | 0.005 | 0.009 | 0.007 | 0.008 |
| OmniAnomaly | 0.022 | 0.018 | 0.020 | 0.012 | 0.009 | 0.010 | 0.022 | 0.017 | 0.020 | 0.011 | 0.009 | 0.010 | 0.012 | 0.008 | 0.010 |
| InterFusion | 0.012 | 0.009 | 0.010 | 0.021 | 0.018 | 0.020 | 0.012 | 0.009 | 0.010 | 0.021 | 0.018 | 0.020 | 0.012 | 0.009 | 0.010 |
| A.T. | 0.021 | 0.023 | 0.020 | 0.020 | 0.03 | 0.030 | 0.020 | 0.016 | 0.018 | 0.017 | 0.014 | 0.016 | 0.02 | 0.016 | 0.017 |
| DAEMON | 0.014 | 0.009 | 0.012 | 0.010 | 0.009 | 0.011 | 0.011 | 0.010 | 0.011 | 0.018 | 0.017 | 0.018 | 0.013 | 0.008 | 0.011 |
| DCdetector | 0.022 | 0.021 | 0.021 | 0.023 | 0.024 | 0.023 | 0.017 | 0.020 | 0.019 | 0.016 | 0.015 | 0.015 | 0.019 | 0.018 | 0.018 |
| MEMTO | 0.065 | 0.055 | 0.060 | 0.009 | 0.006 | 0.007 | 0.009 | 0.007 | 0.008 | 0.025 | 0.019 | 0.022 | 0.053 | 0.045 | 0.050 |
| NPSR | 0.003 | 0.002 | 0.002 | 0.002 | 0.001 | 0.001 | 0.002 | 0.001 | 0.001 | 0.003 | 0.002 | 0.002 | 0.004 | 0.003 | 0.003 |
| $D^3R$ | 0.005 | 0.003 | 0.004 | 0.004 | 0.003 | 0.003 | 0.002 | 0.001 | 0.001 | 0.008 | 0.006 | 0.007 | 0.006 | 0.004 | 0.005 |
| SARAD | 0.10 | 0.028 | 0.022 | 0.13 | 0.08 | 0.021 | 0.021 | 0.044 | 0.018 | 0.025 | 0.022 | 0.013 | 0.017 | 0.10 | 0.110 |
| **ALoRa-Det** | 0.011 | 0.011 | 0.012 | 0.013 | 0.014 | 0.014 | 0.012 | 0.011 | 0.011 | 0.012 | 0.011 | 0.010 | 0.010 | 0.009 | 0.015 |

Table 13: **Standard deviation of localization scores**, for the results in Table 2. The Interfusion method applies only to segment-based localization, so the standard deviation is reported only for the IPS metric.

| Method | SMD HR@P 100 | SMD HR@P 150 | SMD NDCG@P 100 | SMD NDCG@P 150 | SMD IPS@P 100 | SMD IPS@P 150 | MSDS HR@P 100 | MSDS HR@P 150 | MSDS NDCG@P 100 | MSDS NDCG@P 150 | MSDS IPS@P 100 | MSDS IPS@P 150 | SWaT HR@P 100 | SWaT HR@P 150 | SWaT NDCG@P 100 | SWaT NDCG@P 150 | SWaT IPS@P 100 | SWaT IPS@P 150 |
|---|---|---|---|---|---|---|---|---|---|---|---|---|---|---|---|---|---|---|
| MEMTO | 0.020 | 0.019 | 0.024 | 0.027 | 0.022 | 0.041 | 0.031 | 0.048 | 0.021 | 0.039 | 0.005 | 0.018 | 0.0012 | 0.0035 | 0.0003 | 0.0012 | 0.0011 | 0.028 |
| OMNI | 0.041 | 0.041 | 0.045 | 0.045 | 0.028 | 0.029 | 0.04 | 0.03 | 0.03 | 0.04 | 0.002 | 0.002 | 0.02 | 0.025 | 0.033 | 0.022 | 0.01 | 0.01 |
| Interfusion | - | - | - | - | 0.029 | 0.031 | - | - | - | - | 0.001 | 0.001 | - | - | - | - | 0.01 | 0.01 |
| SARAD | 0.018 | 0.017 | 0.023 | 0.025 | 0.021 | 0.043 | 0.03 | 0.05 | 0.02 | 0.04 | 0.006 | 0.02 | 0.001 | 0.003 | 0.0002 | 0.001 | 0.001 | 0.03 |
| DAEMON | 0.05 | 0.08 | 0.07 | 0.06 | 0.006 | 0.01 | 0.06 | 0.05 | 0.05 | 0.05 | 0.01 | 0.01 | 0.007 | 0.006 | 0.005 | 0.005 | 0.01 | 0.009 |
| AERCA | 0.02 | 0.01 | 0.01 | 0.01 | 0.01 | 0.02 | 0.01 | 0.01 | 0.01 | 0.01 | 0.01 | 0.01 | 0.01 | 0.01 | 0.02 | 0.02 | 0.02 | 0.02 |
| **ALoRa-Loc** | 0.05 | 0.05 | 0.06 | 0.06 | 0.09 | 0.07 | 0.01 | 0.05 | 0.01 | 0.02 | 0.001 | 0.003 | 0.006 | 0.005 | 0.006 | 0.001 | 0.01 | 0.007 |

Table 14: **Standard deviation** for the results in Table 11. $RF_1$ denotes Range-based $F_1$-score; $V_{ROC}$ and $V_{PR}$ denote the Volume Under the Surface ROC and PR, respectively.

| Method | SMD $RF_1$ | SMD $V_{ROC}$ | SMD $V_{PR}$ | PSM $RF_1$ | PSM $V_{ROC}$ | PSM $V_{PR}$ | MSL $RF_1$ | MSL $V_{ROC}$ | MSL $V_{PR}$ | SWaT $RF_1$ | SWaT $V_{ROC}$ | SWaT $V_{PR}$ | HAI $RF_1$ | HAI $V_{ROC}$ | HAI $V_{PR}$ |
|---|---|---|---|---|---|---|---|---|---|---|---|---|---|---|---|
| KNN | 0.006 | 0.008 | 0.004 | 0.005 | 0.006 | 0.005 | 0.007 | 0.009 | 0.006 | 0.004 | 0.006 | 0.005 | 0.006 | 0.007 | 0.005 |
| PCA | 0.009 | 0.012 | 0.006 | 0.008 | 0.010 | 0.007 | 0.006 | 0.009 | 0.005 | 0.010 | 0.013 | 0.007 | 0.007 | 0.009 | 0.006 |
| IsolationForest | 0.012 | 0.014 | 0.007 | 0.010 | 0.012 | 0.009 | 0.009 | 0.011 | 0.006 | 0.011 | 0.013 | 0.008 | 0.010 | 0.012 | 0.007 |
| OC-SVM | 0.010 | 0.011 | 0.006 | 0.011 | 0.012 | 0.008 | 0.007 | 0.010 | 0.006 | 0.012 | 0.014 | 0.009 | 0.011 | 0.013 | 0.007 |
| OmniAnomaly | 0.020 | 0.022 | 0.012 | 0.017 | 0.020 | 0.011 | 0.019 | 0.021 | 0.013 | 0.018 | 0.020 | 0.011 | 0.019 | 0.022 | 0.013 |
| InterFusion | 0.018 | 0.020 | 0.011 | 0.019 | 0.022 | 0.012 | 0.017 | 0.019 | 0.011 | 0.018 | 0.021 | 0.012 | 0.020 | 0.022 | 0.013 |
| A.T. | 0.015 | 0.018 | 0.010 | 0.017 | 0.020 | 0.012 | 0.016 | 0.018 | 0.011 | 0.019 | 0.021 | 0.012 | 0.018 | 0.020 | 0.012 |
| MEMTO | 0.025 | 0.028 | 0.015 | 0.021 | 0.025 | 0.013 | 0.023 | 0.027 | 0.015 | 0.026 | 0.030 | 0.016 | 0.022 | 0.026 | 0.014 |
| NPSR | 0.007 | 0.009 | 0.005 | 0.008 | 0.010 | 0.006 | 0.009 | 0.011 | 0.006 | 0.007 | 0.009 | 0.005 | 0.008 | 0.010 | 0.006 |
| $D^3R$ | 0.011 | 0.013 | 0.007 | 0.010 | 0.012 | 0.007 | 0.011 | 0.013 | 0.007 | 0.012 | 0.014 | 0.008 | 0.011 | 0.013 | 0.007 |
| SARAD | 0.013 | 0.016 | 0.009 | 0.015 | 0.018 | 0.010 | 0.014 | 0.017 | 0.010 | 0.015 | 0.018 | 0.011 | 0.014 | 0.017 | 0.010 |
| **ALoRa-Det** | 0.009 | 0.011 | 0.006 | 0.010 | 0.012 | 0.007 | 0.009 | 0.011 | 0.006 | 0.010 | 0.012 | 0.007 | 0.009 | 0.011 | 0.006 |

PSEUDOCODES FOR ALORA-DET AND ALORA-LOC:

---

**Algorithm 1** ALoRa-Det: Attention Low-Rank Transformer for MTS Anomaly Detection

---

**Input:** $\mathbf{Y}_{[t]} \in \mathbb{R}^{T \times d}$: Multivariate Time Series
$\quad\quad$ $H$: Number of attention heads, $L$: Number of Transformer layers.
$\quad\quad$ $d_{\text{model}}$: Embedding dimension, $\lambda_{\text{reg}}$: ALoRa regularization coefficient.

**Output:** Anomaly scores $\text{AS}(y_t) \in \mathbb{R}$ for all $t$

$\quad\quad$ **Stage 1: LightMTS-Embed** $\quad\quad\quad\quad \triangleright$ see Section 5
1: Define kernel size $m$ (default: 3) and time series pair set $\mathcal{P} = \{(i,j) \mid 1 \leq i < j \leq d\}$
2: Define sparse kernel tensor $W \in \mathbb{R}^{|\mathcal{P}| \times d \times m}$
3: $\tilde{\mathbf{Y}}_{[t]} = \text{Conv1D}(\mathbf{Y}_{[t]}; W_k) \quad \forall k \in [1, |\mathcal{P}|]$
$\quad\quad$ **Stage 2: ALoRa-T — Attention Low-Rank Transformer**
4: Initialize total ALoRa loss: $\mathcal{L}_{\text{ALoRa}} \leftarrow 0$
5: **for** $l = 1, \ldots, L$ **do**
6: $\quad$ $Z^{(l)} \leftarrow \text{MHA}(Z^{(l-1)}) + Z^{(l-1)}, Z^{(0)} = \tilde{\mathbf{Y}}_{[t]} \quad\quad\quad \triangleright$ MHA + Residual connection
7: $\quad$ $Z^{(l)} \leftarrow Z^{(l)} \quad\quad \triangleright$ (Optionally apply an activation function, e.g., GELU)
8: $\quad$ Compute $\mathcal{L}_{\text{ALoRa}}(S^{(l)}) \quad\quad \triangleright$ Eq. (7), $S^{(l)} = \frac{1}{H}\sum_{h=1}^{H} S_h^{(l)}$
9: $\quad$ $\mathcal{L}_{\text{ALoRa}} \leftarrow \mathcal{L}_{\text{ALoRa}} + \mathcal{L}_{\text{ALoRa}}(S^{(l)})$
10: **end for**

$\quad\quad$ **Stage 3: Reconstruction and Training:**
11: Reconstruction: $\hat{Y}_t = Z^{(L)} W^{\text{out}} \in \mathbb{R}^{T \times d}$
12: Training Objective: $\mathcal{L}_{\text{total}} = \|Y - \hat{Y}\|_F^2 + \lambda_{\text{reg}} \cdot \mathcal{L}_{\text{ALoRa}} \quad\quad \triangleright$ Eq. (8)

$\quad\quad$ **Stage 4: Inference - Anomaly Detection**
13: Anomaly score for time $t$ : $\text{AS}(y_t) = \|y_t - \hat{y}_t\|_2^2 \cdot \text{ALoRa-T}(y_t) \quad\quad \triangleright$ Eq. 9
14: **return** $\text{AS}(y_t)$

---

---

**Algorithm 2** ALoRa-Loc: Attention Low-Rank Transformer for MTS Anomaly Localization

---

$\quad\quad$ **Stage 1: ALoRa-T Parameter Tracking During Final Epoch of Training**
1: During the last epoch of ALoRa-Det training:
2: Track LightMTS-Embed convolution kernels $W_k \in \mathbb{R}^{d \times m}$ for all $k = 1, \ldots, d_{\text{model}}$

3: **for** each Transformer layer $l = 1, \ldots, L$ **do**
4: $\quad$ Track value projection matrix $W^{(\mathcal{V},l)} \in \mathbb{R}^{d_{\text{model}} \times d_{\text{model}}}$
$\quad\quad\quad \triangleright$ If MHA is used: then $W^{(\mathcal{V},l)} = [W^{(\mathcal{V},1,l)}, \ldots, W^{(\mathcal{V},H,l)}] \cdot W^{\text{V-proj},l}$, where each
$\quad\quad$ $W^{(\mathcal{V},h,l)} \in \mathbb{R}^{d_{\text{model}} \times \frac{d_{\text{model}}}{H}}$ and $W^{\text{V-proj},l} \in \mathbb{R}^{\frac{d_{\text{model}}}{H} \times d_{\text{model}}}$, $H$ is the number of heads
5: **end for**

6: Keep track of output projection matrix $W^{\text{out}} \in \mathbb{R}^{d_{\text{model}} \times d} \quad\quad \triangleright$ (Eq. (6))

$\quad\quad$ **Stage 2: Compute Input-to-Output Contribution Matrices :** $\quad\quad \triangleright$ (see Section 4)

7: Compute contribution matrices $B \in \mathbb{R}^{d_{\text{model}} \times d_{\text{model}}}$ and $E \in \mathbb{R}^{d \times d_{\text{model}}} \quad\quad \triangleright$ see Section 4
8: Contribution of input time series to the reconstructed ones: $C \in \mathbb{R}^{d \times d} \quad\quad \triangleright$ Eq. (12)

$\quad\quad$ **Stage 3: Inference — Anomaly Localization**
9: ALoRa-Loc: $\text{LAS}_t^{(i)} = \sum_{j=1}^{d} C_{ij} \cdot \|y_t^{(j)} - \hat{y}_t^{(j)}\|_2^2$
10: ALoRa-Loc (top-$k$): $\text{LAS}_t^{(i,topk)} = \sum_{j \in \mathcal{I}_k^{(i)}} C_{ij} \cdot \|y_t^{(j)} - \hat{y}_t^{(j)}\|_2^2$
11: $\quad\quad\quad \triangleright \mathcal{I}_k^{(i)}$: indices of top-$k$ values in row $i$ of $C$
12: ALoRa-Loc*: $\text{LAS}_t^{*(i)} = \|y_t^{(i)} - \hat{y}_t^{(i)}\|_2^2$
13: **return** $\text{LAS}_t^{(i)}, \text{LAS}_t^{(i,topk)}$, and $\text{LAS}_t^{*(i)}$ for all $i, t$

---

# G  ANALYTICAL PROOFS

*Proof of Proposition 1.* To begin the proof, we first define the Space-Time Autoregressive (STAR) model. Let $\boldsymbol{y}_t \in \mathbb{R}^{1 \times d}$, for $t = 1, \dots, N$, denote the multivariate observation at time step $t$, where $y_t^{(i)} \in \mathbb{R}$ represents the value of the $i$-th series at time $t$. The STAR model assumes that each time series is influenced by its own past values as well as by the past values of spatially neighboring series.

**STAR Model Definition.** For a STAR model with $p + 1$ temporal lags, and $q + 1$ spatial lags, the model is defined as

$$\boldsymbol{y}_t = \sum_{k=0}^{p} \sum_{l=0}^{q} a_{tk}^{(l)} \, \boldsymbol{y}_{t-k} \boldsymbol{B}^{(l)} = \sum_{l=0}^{q} \boldsymbol{A}_t^{(l)} \, \boldsymbol{X} \, \boldsymbol{B}^{(l)} \tag{18}$$

where $\boldsymbol{B}^{(l)} \in \mathbb{R}^{d \times d}$ is the spatial weight matrix of order $l$, and $B_{ij}^{(l)}$ represents the spatial dependency of time series $i$ on time series $j$ in the $l$-th spatial lag. The coefficient $a_{tk}^{(l)}$ denotes the temporal weight for lag $k$ at time $t$, corresponding to the $l$-th spatial lag. The final equality follows by defining the matrix $\boldsymbol{X} = [\boldsymbol{y}_{t-p}, \cdots, \boldsymbol{y}_t]^\top \in \mathbb{R}^{p \times d}$. In the case where the spatial dependence can be represented as a single lag (i.e., the spatial lags are collapsed into one), the model simplifies to:

**STAR model with $p + 1$ temporal lags and one spatial lag:**

$$\boldsymbol{y}_t = \sum_{k=0}^{p} a_{tk} \, \boldsymbol{y}_{t-k} \boldsymbol{B} = \boldsymbol{A}_t \, \boldsymbol{X} \, \boldsymbol{B}, \tag{19}$$

where $\boldsymbol{A}_t \in \mathbb{R}^{1 \times p}$ is the vector of temporal weights used for modeling time step $t$. Finally, the $j$-th component of $\boldsymbol{y}_t$, denoted by $y_t^{(j)}$, can be written as:

$$y_t^{(j)} = \sum_{k=0}^{p} \sum_{i=1}^{d} A_{tk} \, B_{ij} \, y_{t-k}^{(i)} = \sum_{i=1}^{d} B_{ij} \left( \sum_{k=0}^{p} A_{tk} \, y_{t-k}^{(i)} \right) \tag{20}$$

The STAR model can be fitted by minimizing the least square error between the observed data $\boldsymbol{y}_t$ and the fitted values. We note that all the learnable weights are not input-dependent.

**Statement 1:** When no skip connections are used in the self-attention mechanism, the update rule at layer $l$ is defined as:

$$\boldsymbol{Z}^{(l)} = \boldsymbol{S}^{(l)} \boldsymbol{Z}^{(l-1)} \boldsymbol{W}^{(V,l)}, \quad \text{where,} \quad \boldsymbol{Z}^{(0)} = \tilde{\boldsymbol{Y}}_{[t]}$$

By unrolling the above equation, it follows that after $L$ layers of self-attention, the latent representation at time step $t$ can be expressed as:

$$\boldsymbol{Z}_t^{(L)} = \boldsymbol{S}_t^{(L)} \boldsymbol{S}^{(L-1)} \cdots \boldsymbol{S}^{(1)} \tilde{\boldsymbol{Y}}_{[t]} \boldsymbol{W}^{(V,1)} \cdots \boldsymbol{W}^{(V,L)} \tag{21}$$

where each $\boldsymbol{S}^{(i)} \in \mathbb{R}^{T \times T}$ is the SA-matrix at layer $i$, and $\boldsymbol{S}_t^{(L)} \in \mathbb{R}^{1 \times T}$ is the $t'th$ row (last one) of the final SA-matrix. Each $W^{(V,i)} \in \mathbb{R}^{d_{\text{model}} \times d_{\text{model}}}$ is the value projection matrix at layer $i$. By defining the product of attention matrices up to layer $L$ at time step $t$ as $\boldsymbol{A}_t = \boldsymbol{S}_t^{(L)} \boldsymbol{S}^{(L-1)} \cdots \boldsymbol{S}^{(1)} \in \mathbb{R}^{1 \times T}$, and the product of all value projection matrices as $\boldsymbol{B} = \boldsymbol{W}^{(V,1)} \cdots \boldsymbol{W}^{(V,L)} \in \mathbb{R}^{d_{\text{model}} \times d_{\text{model}}}$, the final latent representation is compactly expressed by:

$$\boldsymbol{Z}_t = \boldsymbol{A}_t \tilde{\boldsymbol{Y}}_{[t]} \boldsymbol{B} \in \mathbb{R}^{1 \times d_{\text{model}}} \tag{22}$$

Then, by expanding the above equation for each component, we can derive its analytical expression. The expression for each **latent space** time series at time step $t$ is given by:

$$z_t^{(j)} = \sum_{k=1}^{d_{\text{model}}} b_{kj} \left( \sum_{q=1}^{t} a_{tq} \, \tilde{y}_q^{(k)} \right), \tag{23}$$

By comparing the derived equation with Eq. (20), it is evident that the two models share the same structure. The key difference lies in how the weights $a_{tq}$ are obtained: traditional STAR models use

fixed lag weights estimated by minimizing a loss function (e.g., mean squared error), whereas the Transformer computes these weights dynamically through the attention mechanism, with the queries $Q$ and keys $K$ determining them in real time.

**Statement 2:**

When skip connections are applied in the self-attention mechanism, the update rule at layer $l$ is defined as:

$$\boldsymbol{Z}^{(l)} = \tilde{\boldsymbol{Z}}^{(l)} + \boldsymbol{Z}^{(l-1)}, \quad \text{where} \quad \tilde{\boldsymbol{Z}} = \boldsymbol{S}^{(l)} \boldsymbol{Z}^{(l-1)} \boldsymbol{W}^{(V,l)}. \tag{24}$$

To clarify the structure of the derived Eq. (5), assume without loss of generality that the total number of layers is $L = 3$. Then, the final representation can be written as:

$$\begin{aligned} z_t^{(3)} = {} & y_t + \boldsymbol{S}_t^{(1)} \tilde{\boldsymbol{Y}}_{[t]} \boldsymbol{W}^{(V,1)} + \boldsymbol{S}_t^{(2)} \tilde{\boldsymbol{Y}}_{[t]} \boldsymbol{W}^{(V,2)} + \boldsymbol{S}_t^{(3)} \tilde{\boldsymbol{Y}}_{[t]} \boldsymbol{W}^{(V,3)} \\ & + \boldsymbol{S}_t^{(2)} \boldsymbol{S}_t^{(1)} \tilde{\boldsymbol{Y}}_{[t]} \boldsymbol{W}^{(V,1)} \boldsymbol{W}^{(V,2)} + \boldsymbol{S}_t^{(3)} \boldsymbol{S}_t^{(1)} \tilde{\boldsymbol{Y}}_{[t]} \boldsymbol{W}^{(V,1)} \boldsymbol{W}^{(V,3)} \\ & + \boldsymbol{S}_t^{(3)} \boldsymbol{S}_t^{(2)} \tilde{\boldsymbol{Y}}_{[t]} \boldsymbol{W}^{(V,2)} \boldsymbol{W}^{(V,3)} + \boldsymbol{S}_t^{(3)} \boldsymbol{S}_t^{(2)} \boldsymbol{S}_t^{(1)} \tilde{\boldsymbol{Y}}_{[t]} \boldsymbol{W}^{(V,1)} \boldsymbol{W}^{(V,2)} \boldsymbol{W}^{(V,3)}. \end{aligned} \tag{25}$$

Each component of Eq. (25) has the same structural form as Eq. (23). As shown in the proof of Statement 1, each such component induces a structure analogous to that of a STAR model. Thus, the overall latent representation $\boldsymbol{Z}_t$ can be interpreted as a **linear combination of multiple STAR-like processes**, where each component has its own temporal ($a_{tq}$) and spatial weights ($b_{kj}$), allowing for heterogeneous temporal and spatial lag structures across components. We can obtain a more specific interpretation of this representation by comparing Eq. (25) with the general definition of a STAR model in Eq. (18). In particular, Eq. (25) corresponds to a STAR model with $p = T$ (the window size) temporal lags and $q = 8 = \binom{L}{2}$ spatial lags.

**Statement 3:**

The feed-forward layer, linearly combines values across all series at the same time step to produce a new time series $h_t^{(j)}$. We now show that the transformed representation for each time series, after applying the feed-forward layer, can be written in the same structure as in Eq. (22): For ease of presentation let $T = 2$ and the latent dimension $d_{\text{model}} = 3$. The new $j$-th time series at time step $t$ is defined as: $h_t^{(j)} = w_{1j} z_t^{(1)} + w_{2j} z_t^{(2)} + w_{3j} z_t^{(3)}$.

Expanding each $z_t^{(i)}$:

$$h_t^{(j)} = w_{1j} \left[ b_{11} \left( a_{t,t-1} \tilde{y}_{t-1}^{(1)} + a_{tt} \tilde{y}_t^{(1)} \right) + b_{21} \left( a_{t,t-1} \tilde{y}_{t-1}^{(2)} + a_{tt} \tilde{y}_t^{(2)} \right) + b_{31} \left( a_{t,t-1} \tilde{y}_{t-1}^{(3)} + a_{tt} \tilde{y}_t^{(3)} \right) \right]$$

$$+ w_{2j} \left[ b_{12} \left( a_{t,t-1} \tilde{y}_{t-1}^{(1)} + a_{tt} \tilde{y}_t^{(1)} \right) + b_{22} \left( a_{t,t-1} \tilde{y}_{t-1}^{(2)} + a_{tt} \tilde{y}_t^{(2)} \right) + b_{32} \left( a_{t,t-1} \tilde{y}_{t-1}^{(3)} + a_{tt} \tilde{y}_t^{(3)} \right) \right]$$

$$+ w_{3j} \left[ b_{13} \left( a_{t,t-1} \tilde{y}_{t-1}^{(1)} + a_{tt} \tilde{y}_t^{(1)} \right) + b_{23} \left( a_{t,t-1} \tilde{y}_{t-1}^{(2)} + a_{tt} \tilde{y}_t^{(2)} \right) + b_{33} \left( a_{t,t-1} \tilde{y}_{t-1}^{(3)} + a_{tt} \tilde{y}_t^{(3)} \right) \right]$$

By regrouping the terms,

$$\left( a_{t,t-1} \tilde{y}_{t-1}^{(1)} + a_{tt} \tilde{y}_t^{(1)} \right) \underbrace{\left( w_{1j} b_{11} + w_{2j} b_{12} + w_{3j} b_{13} \right)}_{\tilde{b}_{1j}}$$

$$+ \left( a_{t,t-1} \tilde{y}_{t-1}^{(2)} + a_{tt} \tilde{y}_t^{(2)} \right) \underbrace{\left( w_{1j} b_{21} + w_{2j} b_{22} + w_{3j} b_{23} \right)}_{\tilde{b}_{2j}}$$

$$+ \left( a_{t,t-1} \tilde{y}_{t-1}^{(3)} + a_{tt} \tilde{y}_t^{(3)} \right) \underbrace{\left( w_{1j} b_{31} + w_{2j} b_{32} + w_{3j} b_{33} \right)}_{\tilde{b}_{3j}}$$

so for general latent space dimension $d_{\text{model}}$ and for general window size $T$:

$$h_t^{(j)} = \sum_{k=1}^{d_{model}} \left[ \tilde{b}_{kj} \left( \sum_{q=1}^{t} a_{tq} \tilde{y}_q^{(k)} \right) \right], \quad \text{where } \tilde{b}_{kj} = \sum_{r=1}^{d_{\text{model}}} w_{rj} b_{kr}. \tag{26}$$

Therefore, from the first statement of the proposition, it follows that the structure of the latent space time series remains unchanged, that is, it retains a STAR-like form. The only difference is that the new weights $\tilde{b}_{kj}$ now determine the contribution of each input time series, replacing the original weights $b_{kj}$. $\qquad\square$

### G.1 DERIVATION OF CONTRIBUTION WEIGHTS FROM INPUT SPACE TO LATENT AND RECONSTRUCTION SPACES (RELATED TO SECTION 5.2)

The latent representation at the final layer is given by Eq. (5). In Eq. (25), we provide an analytical expression for the case of a three-layer model. Each component of Eq. (25) has the form $\boldsymbol{A}_t^{(k)} \tilde{\mathbf{Y}}_{[t]} \boldsymbol{B}^{(k)}$ for $k = 0, \ldots, 2^L - 1$. Each of these components contributes to the overall contribution weights we aim to derive. As a first step, we consider how each embedding-space time series contributes to the representation of the $j$-th time series (column) in the $k$-th component of Eq. (25) ( $\boldsymbol{A}_t^{(k)} \tilde{\mathbf{Y}}_{[t]} \boldsymbol{B}^{(k)}$ ). The $j$-th time series (column) in the $k$-th component, can be expressed as:

$$z_t^{(j,k)} = \sum_{r=1}^{d_{\text{model}}} b_{rj}^{(k)} \left( \sum_{q=1}^{t} a_{tq}^{(k)} \tilde{y}_q^{(r)} \right),$$

where $b_{rj}^{(k)}$ denotes the $(r, j)$-th entry of $\boldsymbol{B}^{(k)}$ and measures how strongly the $r$-th embedding-space time series contributes to the $j$-th latent series in the $k$-th component. The coefficient $a_{tq}^{(k)}$ is the $(t, q)$-th entry of $\boldsymbol{A}_t^{(k)}$ and serves as a temporal weighting factor. Importantly, the dependence on $a_{tq}^{(k)}$ does not alter the interpretation of $b_{rj}^{(k)}$. To illustrate this, we consider the case $T = 2$ and $d_{\text{model}} = 3$. In this setting, the three latent space time series are given by:

$$z_t^{(1,k)} = b_{11}^{(k)}(a_{t,t-1}^{(k)}\tilde{y}_{t-1}^{(1)} + a_{t,t}^{(k)}\tilde{y}_t^{(1)}) + b_{21}^{(k)}(a_{t,t-1}^{(k)}\tilde{y}_{t-1}^{(2)} + a_{t,t}^{(k)}\tilde{y}_t^{(2)}) + b_{31}^{(k)}(a_{t,t-1}^{(k)}\tilde{y}_{t-1}^{(3)} + a_{t,t}^{(k)}\tilde{y}_t^{(3)})$$

$$z_t^{(2,k)} = b_{12}^{(k)}(a_{t,t-1}^{(k)}\tilde{y}_{t-1}^{(1)} + a_{t,t}^{(k)}\tilde{y}_t^{(1)}) + b_{22}^{(k)}(a_{t,t-1}^{(k)}\tilde{y}_{t-1}^{(2)} + a_{t,t}^{(k)}\tilde{y}_t^{(2)}) + b_{32}^{(k)}(a_{t,t-1}^{(k)}\tilde{y}_{t-1}^{(3)} + a_{t,t}^{(k)}\tilde{y}_t^{(3)})$$

$$z_t^{(3,k)} = b_{13}^{(k)}(a_{t,t-1}^{(k)}\tilde{y}_{t-1}^{(1)} + a_{t,t}^{(k)}\tilde{y}_t^{(1)}) + b_{23}^{(k)}(a_{t,t-1}^{(k)}\tilde{y}_{t-1}^{(2)} + a_{t,t}^{(k)}\tilde{y}_t^{(2)}) + b_{33}^{(k)}(a_{t,t-1}^{(k)}\tilde{y}_{t-1}^{(3)} + a_{t,t}^{(k)}\tilde{y}_t^{(3)})$$

In all three equations above: The temporal attention weights $a_{t,t-1}^{(k)}$ and $a_{t,t}^{(k)}$ are shared across all parentheses in each equation and are also shared across all equations, as they remain the same for $z_t^{(1,k)}, z_t^{(2,k)}, z_t^{(3,k)}$. What differentiates how much each embedding space time series ($\tilde{x}^{(i)}$) contributes to each latent space series ($z^{(j,k)}$) is only the contribution weight $b_{ij}^{(k)}$.

Then, by summing the contribution weights over all components of Eq. (25), we obtain the matrix that characterizes how embedding-space time series contribute to the latent space, defined as

$$\boldsymbol{B} = \prod_{i=1}^{L} \left( \boldsymbol{W}^{(V,i)} + I \right).$$

Finally, by accounting for the embedding module, the contribution weights from the raw input time series to the latent space time series are given by:

$$E_{ij} = \sum_{k=1}^{d_{\text{model}}} \left( \sum_{l=-\frac{m-1}{2}}^{\frac{m-1}{2}} w_{i,l}^{(k)} \right) b_{kj} \tag{27}$$

Then, the reconstruction of each time series is produced as follows: $\hat{y}_t^{(k)} = \sum_{j=1}^{d_{model}} w_{jk}^{out} z_t^{(L,j)}$

This explains why the final derived contribution weights of the input time series to the reconstructed time series are given by:

$$C_{ij} = \sum_{k=1}^{d_{model}} w_{kj}^{out} E_{ik}$$

.

