# OpenReview forum: "Low Rank Transformer for Multivariate Time Series Anomaly Detection and Localization"
_ICLR.cc/2026/Conference — ICLR 2026 Poster_

### Official Review · Reviewer_Qfvm · 2025-10-29

**Soundness:** 4
**Presentation:** 4
**Contribution:** 3
**Rating:** 6
**Confidence:** 5

**Summary:**

The paper tackles multivariate time series anomaly diagnosis, covering both detection and localization. It analyzes the learning behavior of Transformers from a theoretical perspective and connects it to classical statistical time-series analysis. Based on these insights, the authors propose the Attention Low-Rank Transformer (ALoRa-T) with low-rank regularization to better capture temporal anomaly patterns, and introduce ALoRa-Loc for variable-level anomaly localization. Experiments on real and synthetic datasets show that the proposed approach outperforms existing methods in both detection and localization tasks.

**Strengths:**

1. The paper offers valuable theoretical insights by linking the Transformer’s self-attention mechanism to established statistical time-series principles, providing a more interpretable foundation for deep anomaly detection models.
2. Unlike many prior works focusing only on detection, the introduction of ALoRa-Loc enables variable-level anomaly attribution, advancing the underexplored area of multivariate anomaly localization.

**Weaknesses:**

1. The distinction between “time series” and “variable” is not consistently maintained throughout the paper. Since each variable corresponds to a univariate time series, the terminology should be clarified to avoid conceptual confusion.
2. The paper states that each kernel learns representations from only two time series, but the motivation for selecting exactly two is not explained.
3. The metrics used to evaluate anomaly localization ability — Hit Rate, Normalized Discounted Cumulative Gain (NDCG), and Interpretation Score — are not well-suited for this task. Hit Rate and NDCG are designed for ranking or recommendation settings, while Interpretation Score lacks a clear definition in the context of anomaly localization.
4. The paper does not compare with established approaches for anomaly localization or root cause identification, such as “Root Cause Analysis of Anomalies in Multivariate Time Series through Granger Causal Discovery.”
5. In Table 2, several numeric values use commas instead of decimal points.

**Questions:**

Please see the weaknesses.

---

> ### Author Response · Authors · 2025-11-24
>
> We thank the Reviewer for his/her in-depth review of our work and for providing us with very insightful comments that have helped us improve the presentation and the quality of the paper.
>
>
>
> ## Reply to Weaknesses:
>
> ### Reply to Weakness 1:
>
> We thank the Reviewer for this comment, which helped us improve the clarity of our paper. To address this, we revised the terminology throughout the paper to ensure consistent use of “time series” instead of “variable” in all places where confusion could arise. We only retain the term “variable” where it is strictly necessary. For example, see the first line of Section 3 (MTS notation) and a sentence in the related work where “variable” is the appropriate term based on the context.
>
> ### Reply to Weakness 2:
> Thank you for allowing us to clarify this. The proposed LightMTS embedding module was designed in such a way as to reduce complexity. Specifically, increasing the number of time series involved in each interaction, from pairs to triplets or higher-order groups, results in a **combinatorial explosion** in the number of possible interactions. In such cases, the complexity becomes comparable to using standard fully connected or convolutional embeddings. Formally, the number of combinations grows according to the binomial coefficient $\binom{D}{k} = O(D^k)$ for fixed k.
>
> This information has been added to the revised paper in Appendix D.1, specifically in the new paragraph titled _“Why LightMTS embedding module use pairwise interactions.”_
>
> ### Reply to Weakness 3:
> We agree with the Reviewer’s comment that the original version of these metrics were introduced for recommendation settings. However, they have since been extended, **and are now widely used for addressing the task of MTS localization** [R1,R2,R3,R4] as suitable evaluation metrics.
>
> ### Reply to Weakness 4:
> To address this comment, we followed the Reviewer’s recommendation and included the suggested methodology (AERCA ), introduced by Han et al. [R6] in our evaluation, as it is one of the most recent state-of-the-art methods for the localization task. Additionally, we have included another baseline method (DAEMON) to further strengthen the localization evaluation. **The corresponding results can be found in Table 2 of the revised paper, where our proposed method still significantly outperform, in most cases these methods.**
>
>
> ### Reply to Weakness 5:
> We thank the Reviewer for identifying these issues and helping us improve the presentation of our paper. In the revised version, this issue has been addressed.
>
>
>
>
> ## References
> [R1] Zhihao Dai, Ligang He, Shuanghua Yang, and Matthew Leeke. Sarad: Spatial association-aware anomaly detection and diagnosis for multivariate time series. In Advances in Neural Information Processing Systems (NeurIPS), 2024. URL https://openreview.net/forumid= gmf5Aj01Hz.
> [R2] Shreshth Tuli, Giuliano Casale, and Nicholas R. Jennings. 2022. TranAD: deep transformer networks for anomaly detection in multivariate time series data. Proc. VLDB Endow. 15, 6 (feb 2022), 1201–1214. https://doi.org/10.14778/3514061.3514067
>
> [R3] Hang Zhao, Yujing Wang, Juanyong Duan, Congrui Huang, Defu Cao, Yunhai Tong, Bixiong Xu, Jing Bai, Jie Tong, and Qi Zhang. 2020. Multivariate Time-Series Anomaly Detection via Graph Attention Network. In 2020 IEEE International Conference on Data Mining (ICDM). 841–850. https://doi.org/10.1109/ICDM50108.2020.00093
>
> [R4] Zhihan Li, Youjian Zhao, Jiaqi Han, Ya Su, Rui Jiao, Xidao Wen, and Dan Pei. Multivariate Time Series Anomaly Detection and Interpretation using Hierarchical Inter-Metric and Temporal Embedding. In Proceedings of the 27th ACM SIGKDD Conference on Knowledge Discovery & Data Mining, pp. 3220–3230. ACM, August 2021. ISBN 978-1-4503-8332-5. doi: 10.1145/3447548.3467075.
>
> [R5]Y. Su, Y. Zhao, C. Niu, R. Liu, W. Sun, and D. Pei, “Robust anomaly detection for multivariate time series through stochastic recurrent neural network,” in Proceedings of the 25th ACM SIGKDD International Conference on Knowledge Discovery & Data Mining, 2019, pp. 2828– 2837.
>
> [R6]Xiao Han, Saima Absar, Lu Zhang, and Shuhan Yuan. Root cause analysis of anomalies in multivariate time series through granger causal discovery. In The Thirteenth International Conference on Learning Representations, 2025. URL https://openreview.net/forumid= k38Th3x4d9.

---

### Official Review · Reviewer_NVbX · 2025-10-30

**Soundness:** 2
**Presentation:** 2
**Contribution:** 2
**Rating:** 4
**Confidence:** 4

**Summary:**

This paper proposes a transformer-based framework for time-series anomaly detection that leverages attention rank analysis to interpret and localize anomalies. The key idea is that the rank of self-attention matrices increases when anomalies occur, providing a new signal for both detection and localization.

**Strengths:**

1. The idea of detecting anomalies by analyzing the transformer’s learning behavior is original and insightful. It opens a new direction for understanding model-internal representations in time-series anomaly detection.

2. The focus on anomaly localization is meaningful and practically valuable.

**Weaknesses:**

1. The paper uses Spearman correlation to estimate dependencies among sequence pairs but does not justify why this choice is preferred over Pearson correlation or Cosine Similarity. Furthermore, the paper states that only the top-K correlated pairs are retained, yet the criterion for determining K is not specified or experimentally analyzed.

2. The central claim that “the rank of SA-matrices increases in the presence of anomalies” is only supported by empirical observation on a few datasets. The paper does not provide a theoretical explanation or evidence that this phenomenon holds consistently across diverse anomaly types and domains.

3. The definitions of variables are inconsistent—sometimes the input sequence is denoted as x, other times as y, making the mathematical expressions difficult to follow.

4. The inference process depends critically on the threshold h_2. Although the paper mentions that Appendix A describes its selection, the appendix does not include such details yet.

5. Localization evaluation requires ground-truth information about the precise anomalous series. However, the datasets used in the experiments typically provide only record-level anomaly labels (anomalous or normal per timestamp) without explicit localization annotations. Could the authors clarify how the localization ground truth is obtained?

6. The main text experiments are overly concise and lack detailed analysis. Although the appendix includes an ablation study, it only evaluates the embedding module. A more critical ablation, particularly on the ALoRa loss function, is missing and should be included to support the claimed effectiveness of the proposed loss.

7. The paper does not provide the source code, and the methodological descriptions are not detailed enough to reproduce the reported results reliably.

**Questions:**

See the weaknesses section

---

> ### Author Response · Authors · 2025-11-24
>
> We thank the Reviewer for his/her in-depth review of our work and for providing us with very insightful comments that have helped us improve the presentation and the quality of the paper.
>
> ## Reply to Weakness:
> ### Reply to Weakness 1:
> We thank the Reviewer for this comment, which allowed us to clarify the use of Spearman correlation over other alternatives.
>
> Spearman correlation is a robust measure of correlation because it can capture monotonic relationships between variables, while Pearson correlation measures linear relationships. To further support our choice, **we have added an additional ablation study** comparing model performance when using Spearman correlation versus Pearson correlation within the LightMTS-Embed module. This ablation study has been added to the **revised paper (Section 7, Table 4)**, and its results confirm our justification.
>
> **Top-K pairs selection:** To address this comment,
>
> 1.  **We have added an ablation study, presented in the new Table 4 in Section 7**, comparing performance when using all variable pairs versus using only the top-$K$ pairs, where $K = 512$.
> **Conclusion**: Selecting the Top-K pairs substantially reduces computational cost without sacrificing accuracy, and even improves it in some cases, demonstrating the effectiveness of this design.
>
> 2. Additionally, to analyze this choice in depth, we add a paragraph in **Appendix D1** entitled **Top-K Selection for the LightMTS-Embed Module**, which explains the choice of setting $K = 512$ and evaluates the model’s sensitivity to $K$; see **Table 8** of the revised paper.
> **Conclusion**: The results indicate that the model is not sensitive to this choice.  Since our model is already the most computationally efficient, as demonstrated in the original paper (Appendix C), and its computational cost is already very low, making it suitable for real-world applications, further reducing $K$ deems unnecessary.
>
> ### Reply to Weakness 2:
> Providing a formal theoretical guarantee for the intuition that anomalous windows yield higher attention rank is indeed challenging. The main difficulty arises from the unknown and highly variable dynamics of real-world datasets. Any theoretical result would depend on assumptions about the underlying distribution and the structure of “normal” operational behavior, yet these characteristics vary substantially across different domains.
>
> To support this intuition empirically, we included four figures in the **original paper** showing how the attention rank, and consequently the ALoRa-T score, increases when anomalies occur across various systems from different domains: **Figures 3, 4, 10 (Figure 11 in the revised version), and 11 (Figure 12 in the revised version).**
>
> In the revised manuscript, **we further strengthen this evidence** by adding a dedicated paragraph in **Appendix F entitled "_Effectiveness of the ALoRa-T Score Across Anomaly Types"_**.  In this paragraph, we show how the self-attention rank, and consequently the corresponding ALoRa-T score, behaves under different anomaly types, as illustrated in the **new Figure 10 of the revised paper**. Across all anomaly types, both the attention rank and the ALoRa-T score consistently increase in the presence of anomalies.
>
> We have also included an **ablation study on the effectiveness of the ALoRa-T score in the new Table 3 in Section 7**, which demonstrates consistent performance improvements across all datasets. Since these datasets come from diverse application domains, this provides strong empirical support for our intuition.
>
>
> ### Reply to Weakness 3:
> We thank the Reviewer for this helpful comment, which allowed us to improve the clarity of the paper. In response, we have revised the notation throughout the paper to ensure full consistency. Specifically, we now use the symbol **$\mathbf{Y}$** exclusively (and no longer $\mathbf{X}$ ) when referring to the input sequence.
>
> All corresponding changes have been made throughout the revised paper to ensure consistent and clear notation.

---

> > ### Author Response · Authors · 2025-11-24
> >
> > ### Reply to Weakness 4:
> >  We thank the Reviewer for this comment, which helped us clarify the selection of $h_2$ and include it in the revised paper.
> >
> > We would like to clarify that all anomaly detection models, either explicitly or implicitly, **require** a threshold, $h_2$, to convert anomaly scores into binary labels. To avoid introducing bias from any particular threshold-selection strategy and to ensure a **fair comparison across all methods**, we follow a well-established evaluation protocol. For each method, including ALoRa-Det, we report performance using the **best achievable threshold**, i.e., the threshold that yields the best performance in terms of $F_1$ score. This evaluation strategy is widely used in the anomaly detection literature. For example, several recent methods, including **SARAD (NeurIPS 2024)** and **NPSR (NeurIPS 2023)**, follow the same protocol for selecting the threshold $h_2$. Similarly, methods that rely on a specific threshold-selection algorithm such as **SPOT** also report the **best achievable performance** under that procedure (e.x D3R method, NeurIPS 2023), in order to avoid introducing bias from the choice of SPOT hyperparameters. This is the same approach as we take. In summary, this selection strategy ensures **fairness** in comparing all baselines: it evaluates the **quality of the anomaly scores produced by each method**, rather than the particular threshold-selection mechanism used to convert those scores into labels.
> >
> > A corresponding explanation for the $h_2$ selection, consistent with the description above, is **provided in the revised paper in Appendix A**.
> >
> >
> >  ### Reply to Weakness 5:
> > Since anomalies do not occur at the precise physical location of a single sensor, it is generally unrealistic to assume that one specific sensor is “the anomalous one,” except in very simple or fully controlled scenarios. Thus, **MTS localization** aims to **identify** the **time-series** signals whose corresponding physical components were **involved in producing the anomaly.**
> >
> > By:
> > 1. identifying the time-series signals that were most involved in the anomalous event, and
> > 2. providing an explanation of how the system behaves, an aspect that our method addresses (see Sections 4 and 5.2)
> >
> > practitioners have the essential information needed to properly analyze and understand the event.
> >
> > The proposed **ALoRa-Loc** method addresses both of these crucial aspects: **(1)** MTS localization by tracing anomaly to their true source and **(2)** interpretability of the model’s behavior. Both capabilities remain largely underexplored in the research community focused on multivariate time-series anomaly diagnosis.
> >
> >
> > **Regarding how localization labels are obtained in the evaluation datasets:**
> > 1. **Details and information on how each dataset was obtained is given in Appendix B**.
> > 2. SMD dataset: This dataset was introduced in [R1]. Experts used incident reports to label the time series involved in each anomalous event.
> > 3. MSDS: Introduced in [R2]. Localization labels were derived from official reports describing the cause of each anomaly; all annotated time series correspond to components directly involved in the event.
> > 4.  SWaT:  This dataset is available upon request from the well-recognized _"iTrust Center for Research in Cyber Security"_. The SWaT dataset comes from a realistic testbed where controlled anomalies were introduced. Each anomaly was triggered through specific sensors, and only those sensors were labeled as anomalous.

---

> > > ### Author Response · Authors · 2025-11-24
> > >
> > > ### Reply to Weakness 6:
> > > We thank the Reviewer for this comment, which helped us provide additional information to support further our work. **In the revised version**, we have included an **ablation study on the effectiveness of ALoRa-Loss and ALoRa-Score**, along with four additional ablation studies to verify the effectiveness of the individual components of our method. Specifically, we have added:
> > >
> > > **Main paper, Section 7 of the revised paper ( _Ablation studies_)**
> > > 1. **Ablation study on the effectiveness of ALoRa-Loss and ALoRa-Score:** Table 3 of the revised paper.
> > >
> > > **Conclusion**: The **results** demonstrate the **effectiveness of both the ALoRA loss and the ALoRA-T score** in improving anomaly detection across all datasets. This highlights the robustness and general applicability of these components.
> > >
> > > 2. **Extended ablation study on the LightMTS-Embed module**: We added **Table 4**  in Section 7, which reports the selection of Top-K pairs versus using all pairs, and compares Spearman correlation with Pearson correlation.
> > >
> > > **Conclusion**: See our answer to Weakness 1
> > >
> > >
> > > **Appendix D of the revised paper (_Ablation studies supplement_)**
> > >
> > > 3. **Ablation Study on the Top-K pairs selection: The new Table 8** illustrates the sensitivity of the parameter $K$ for the LightMTS-Embed module.
> > >
> > > **Conclusion**: See our answer to Weakness 1
> > >
> > > 4. **Ablation study supporting the decision to omit feed-forward layers**:
> > > In the original paper, the decision to omit feed-forward layers was theoretically motivated by Proposition 1 (statement 3), which shows that these layers do not alter the latent space structure. To further support this claim, we have added an ablation study evaluating both performance and computational cost when feed-forward layers are used and when they are omitted. **This study is presented in Appendix D.2  and summarized in Table 9 of the revised paper**.
> > >
> > > **Conclusion**: The experimental results in Table 9 confirm the theoretical insight. Specifically, when feed-forward layers are included, performance either decreases or remains unchanged, while computational cost increases substantially.
> > >
> > > 5. **Ablation study on the placement of the ALoRa-Loss**, comparing head-wise application versus averaging across self-attention heads;
> > > See  **Appendix D.3** and **Table 10** of the revised paper.
> > >
> > > **Conclusion**: Both variants achieve nearly identical detection performance across all datasets. However, applying the ALoRA loss to the average self-attention matrix is slightly more computationally efficient, and since there is no other advantage to the alternative, this variant was preferred.
> > >
> > >
> > > ### Reply to Weakness 7:
> > > In the original submission the source code **was provided as supplementary material**, together with a detailed README explaining how to reproduce all reported results. In addition, we had included a complete **Reproducibility Statement**, which is highly recommended by ICLR 2026, following the official guidelines.
> > >
> > >
> > >
> > > ## References
> > > [R1] Su et al. , Robust Anomaly Detection for Multivariate Time Series through Stochastic Recurrent Neural Network. ACM SIGKDD , 2019.
> > >
> > > [R2] Nedelkoski et al. ,  Multi-source Distributed System Data for AI-Powered Analytics, ESOCC 2020.

---

> > > > ### Comment · Reviewer_NVbX · 2025-11-28
> > > >
> > > > I appreciate the clarity and effort you put into the explanations. Your responses have fully addressed all of my concerns. If possible, I would be happy to raise my score.

---

### Official Review · Reviewer_2urD · 2025-11-01

**Soundness:** 3
**Presentation:** 3
**Contribution:** 3
**Rating:** 6
**Confidence:** 4

**Summary:**

The paper proposes ALoRa, a Transformer-based framework for multivariate time-series (MTS) anomaly detection and localization grounded in a theoretical analysis of Transformer encoders on MTS. The authors show that the encoder’s latent representations can be expressed as linear combinations of Space-Time Autoregressive (STAR) processes, which motivates (i) ALoRa-T—a Transformer with low-rank regularization on self-attention—and (ii) a detection score that counts significant singular values of the final attention matrix. They further derive contribution weights from inputs → latent → outputs to trace anomaly propagation and attribute anomalies to variables (ALoRa-Loc).

**Strengths:**

(1) The paper provides a coherent spectral perspective on attention that is simple to compute conceptually and ties to an interpretable diagnostic.

(2) The authors diagnose that point-adjustment inflates results—sometimes making them indistinguishable from random scoring—and therefore pivot to range-aware/affiliation-based metrics, improving evaluation validity.

(3) The localization section explicitly models propagation via contribution weights (E, C), which is more principled than per-dimension reconstruction heuristics.

(4) The training objective is compact and implementable; the regularizer integrates cleanly with standard reconstruction losses.

**Weaknesses:**

(1) While the detection pipeline uses two thresholds ($h_1, h_2$), Appendix A provides data-driven approach of choosing threshold $h_1$, but this is still a per-dataset manual step, introducing hyperparameter sensitivity. Also, neither ablation on $h_2$ selection nor heuristics on choosing it was provided.

(2) The paper’s central intuition—\textit{anomalous windows yield higher attention rank}—is supported empirically (plots/observations) but lacks a formal guarantee. No theoretical background specifies conditions under which anomalies must raise rank (or non-anomalies must not).

(3) ALoRa-Loc traces propagated influence, but ranking metrics like HR/NDCG/IPS do not distinguish origin variable from downstream affected variables; without per-segment confusion analyses, it’s unclear whether the method finds causes or merely effects.

**Questions:**

(1) How sensitive is the final detection F1-score (which relies on the combined $AS(x_t)$) to this choice? For instance, what is the performance impact if $h_1$ is set 10x larger or 10x smaller than the value chosen via the eigenvalue distribution analysis?

(2) Please provide explanation on how $h_2$ value was selected and why.

(3) Do different anomaly types (point vs collective vs contextual) induce distinct singular-value patterns? Any class-wise analysis of detection latencies?

(4) For segments where the anomaly propagates widely, how often does top-k ALoRa-Loc identify the true origin vs “most affected” variables? Could you report per-segment confusion analyses?

(5) Some important ablations are missing: rank-only score vs error-only vs multiplicative combo; head-wise vs averaged penalty; all-pair vs top-K embeddings; FFN on/off at matched params. Could you please provide ablations on these?

* A minor typo in line 228; "throught" $\rightarrow$ "thought"

---

> ### Author Response · Authors · 2025-11-24
>
> We thank the Reviewer for his/her in-depth review of our work and for providing us with very insightful comments that have helped us improve the presentation and the quality of the paper.
>
> ## Reply to Weaknesses:
> ### Reply to Weakness 1:
> #### Selection of $h_1$:
> Although $h_1$ is based on a given dataset, it is **not** a manually tuned parameter. Instead, we use an **automated, data-driven procedure** to select it, as **it has to** adapt to the data. Datasets arise from different application domains and therefore may exhibit distinct underlying dynamics, which implies that the **spectral properties of the self-attention matrices under normal operation are not the same**. With the proposed procedure, $h_1$ is automatically determined for **any new dataset** based on the spectral properties of self-attention matrices observed under normal operation. Specifically, we examine the **fourth** and **fifth** largest eigenvalues of the self-attention matrices and track their values across the training sequence. The threshold $h_1$ is then automatically chosen as the **maximum value** observed between these two eigenvalue trajectories. This ensures that only the dominant eigenvalues, typically corresponding to a rank of 3–4, are preserved. **The selection process is provided in the original submission in Appendix A, and we further clarify it in the revised version.**
>
> #### Selection of $h_2$:
> We would like to clarify that all anomaly detection model, either explicitly or implicitly, **requires** a threshold, $h_2$, to convert anomaly scores into binary labels. To avoid introducing bias from any particular threshold-selection strategy and to ensure a **fair comparison across all methods**, we follow a well-established evaluation protocol. For each method, including ALoRa-Det, we report performance using the **best achievable threshold**, i.e., the threshold that yields the best performance in terms of $F_1$ score. This evaluation strategy is widely used in the anomaly detection literature. For example, several recent methods, including **SARAD (NeurIPS 2024) [R2]** and **NPSR (NeurIPS 2023) [R3]**, follow the same protocol for selecting the threshold $h_2$. Similarly, methods that rely on a specific threshold-selection algorithm such as **SPOT** [R1] also report the **best achievable performance** under that procedure (e.x D3R method, NeurIPS 2023), in order to avoid introducing bias from the choice of SPOT hyperparameters. This is the same approach as we take. In summary, this selection strategy ensures **fairness** in comparing all baselines: it evaluates the **quality of the anomaly scores produced by each method**, rather than the particular threshold-selection mechanism used to convert those scores into labels.
>
> A corresponding explanation for the $h_2$ selection, consistent with the description above, is **provided in the revised paper in Appendix A**.
>
> ### Reply to Weakness 2:
> Providing a formal theoretical guarantee for the intuition that anomalous windows yield higher attention rank is indeed challenging. The main difficulty arises from the unknown and highly variable dynamics of real-world datasets. Any theoretical result would depend on assumptions about the underlying distribution and the structure of “normal” operational behavior, yet these characteristics vary substantially across domains. This makes a theoretical formulation difficult to establish.
>
> To support this intuition empirically, we include four figures in the original paper showing how the attention rank, and consequently the ALoRa-T score, increases when anomalies occur: Figures 3, 4, 10 (Figure 11 in the revised version), and 11 (Figure 12 in the revised version). In the revised manuscript, we further strengthen this evidence by adding a dedicated paragraph in Appendix F entitled **_Effectiveness of the ALoRa-T Score Across Anomaly Types_**.  In this paragraph, we analyze how the self-attention rank, and consequently the corresponding ALoRa-T score, behaves under different anomaly types, as illustrated in the **new Figure 10  in Appendix F  of the revised paper**. Across all anomaly types, both the attention rank and the ALoRa-T score consistently increase in the presence of anomalies. We have also included an **ablation study on the effectiveness of the ALoRa-T score in the new Table 3 in Section 7**, which demonstrates better and consistent performance across all datasets. Since these datasets come from diverse application domains, this provides strong empirical support for our intuition. We thank the Reviewer for the comment, which helped to improve the paper.

---

> > ### Author Response · Authors · 2025-11-24
> >
> > ### Reply to Weakness 3:
> > Since anomalies do not occur at the precise physical location of a single sensor, it is generally unrealistic to assume that one specific sensor is “the anomalous one,” except in simple or fully controlled scenarios. Thus, **MTS localization** aims to **identify** the **time-series** signals whose corresponding physical components were **involved in producing the anomaly.**
> >
> > The proposed ALoRa-Loc method integrates both of these crucial capabilities: (1) accurate MTS localization by tracing anomalies to their true sources, and (2) interpretability of the model’s behavior. Both of these aspects remain largely underexplored in the research community focused on multivariate time-series anomaly diagnosis. With these capabilities, practitioners gain the essential information needed to properly analyze and understand anomalous events.
> >
> > For this reason, the evaluation metrics **HR**, **NDCG**, and **IPS** are appropriate and are widely used by recent state-of-the-art methods, as they are well recognized for this task [R2, R4, R5, R6].
> >
> >
> >
> > ## Reply to Questions:
> > ### Reply to Question 1:
> > We thank the reviewer for this question, which helps us clarify the sensitivity of the model to this parameter. As shown in the table below, the model’s performance is **not sensitive** to this parameter, even when it is set to a value **10 times larger** or **10 times smaller** than the default. We also refer the reviewer to our response to **Weakness 1**, where we explain the selection of the parameter $h_1$ based on an automated procedure that analyzes the spectral properties of the self-attention matrices under normal operating conditions.
> >
> >
> > | Threshold Setting                         | SMD F1 | PSM F1 | MSL F1 | SWaT F1 | HAI F1 |
> > |-------------------------------------------|--------|--------|--------|---------|--------|
> > | Automatically selected from normal operation spectral properties.           |     0.97   |     0.82   |     0.72   |   0.68      |     0.86   |
> > | 10× larger       |   0.97     |   0.823     |    0.714    |    0.64     |      0.856  |
> > | 10× smaller             |   0.97     |    0.827    |   0.71     |      0.66   |     0.853   |
> >
> >
> > ### Reply to Question 2:  We refer to our answer to Weakness 1
> >
> > ### Reply to Question 3:  We refer to our answer to Weakness 2
> >
> > ### Reply to Question 4:
> > We thank the Reviewer for this comment, which helped us clarify how the ALoRa-Loc top-k variant works. The proposed ALoRa-Loc method not only addresses MTS localization but also yields the matrices **E** and **C** (Eq. 11 and Eq. 12), which quantifies the contribution of each input time series to the reconstruction of each output time series.
> >
> > The ALoRa-Loc top-k variant is designed for cases where only a subset of input time series significantly contributes to the reconstruction of a given output series (i.e., where the corresponding weights $c_{ij}$ are large). It uses this interpretability information to ignore very small contributions (very low $c_{ij}$ weights) during the localization process, thereby focusing only on the most meaningful contributors. Thus, a practitioner can use the interpretability information, together with their domain knowledge, to determine which components are most influential for each time series.

---

> ### Author Response · Authors · 2025-11-24
>
> ### Reply to Question 5:
> We thank the Reviewer for this comment, which helped us provide additional information to support further our work. **In the revised version**, we have included the following ablation studies:
>
> **Main paper, Section 7 of the revised paper ( _Ablation studies_)**
> 1. **Ablation study on the effectiveness of ALoRa-Loss and ALoRa-Score:** Table 3 of the revised paper.
>
> **Conclusion**: The **results** demonstrate the **effectiveness of both the ALoRA loss and the ALoRA-T score** in improving anomaly detection across all datasets. This highlights the robustness and general applicability of these components.
>
> 2. **Extended ablation study on the LightMTS-Embed module**: We added **Table 4**  in Section 7, which reports the selection of Top-K pairs versus using all pairs, and compares Spearman correlation with Pearson correlation.
> **Conclusion**: Selecting the Top-K pairs substantially reduces computational cost without sacrificing accuracy, and even improves it in some cases, demonstrating the effectiveness of this design.
>
>
> **Appendix D of the revised paper (_Ablation studies supplement_)**
>
> 3. **Ablation Study on the Top-K pairs selection: The new Table 8** illustrates the sensitivity of the parameter $K$ for the LightMTS-Embed module.
>
> **Conclusion**: The results indicate that the model is not sensitive to this choice.  Since our model is already the most computationally efficient, as demonstrated in the original paper (Appendix C), and its computational cost is already very low, making it suitable for real-world applications, further reducing $K$ was unnecessary.
>
> 4. **Ablation study supporting the decision to omit feed-forward layers**:
> In the original paper, the decision to omit feed-forward layers was theoretically motivated by Proposition 1 (statement 3), which shows that these layers do not alter the latent space structure. To further support this claim, we have added an ablation study evaluating both performance and computational cost when feed-forward layers are used and when they are omitted. **This study is presented in Appendix D.2  and summarized in Table 9 of the revised paper**.
>
> **Conclusion**: The experimental results in Table 9 confirm the theoretical insight. Specifically, when feed-forward layers are included, performance either decreases or remains unchanged, while computational cost increases substantially.
>
> 5. **Ablation study on the placement of the ALoRa-Loss**, comparing head-wise application versus averaging across self-attention heads;
> See  **Appendix D.3** and **Table 10** of the revised paper.
>
> **Conclusion**: Both variants achieve nearly identical detection performance across all datasets. However, applying the ALoRA loss to the average self-attention matrix is slightly more computationally efficient, and since there is no other advantage to the alternative, this variant was preferred.
>
>
> ## References
> [R1] Siffer et al. , Anomaly detection in streams with extreme value theory, ACM SIGKDD, 2017.
>
> [R2] Dai et al., Spatial association-aware anomaly detection and diagnosis for multivariate time series. NeurIPS, 2024.
>
> [R3]Lai et al., Nominality score conditioned time series anomaly detection by point/sequential reconstruction. NeurIPS, 2023.
>
> [R4] Tuli et al., TranAD: deep transformer networks for anomaly detection in multivariate time series data. Proc. VLDB Endow. 15, 2022.
>
> [R5] Zhao et al., Multivariate Time-Series Anomaly Detection via Graph Attention Network, ICDM, 2020.
>
> [R6] Li et al., Multivariate Time Series Anomaly Detection and Interpretation using Hierarchical Inter-Metric and Temporal Embedding. ACM SIGKDD, 2021.

---

### Official Review · Reviewer_CXw2 · 2025-11-04

**Soundness:** 3
**Presentation:** 2
**Contribution:** 3
**Rating:** 6
**Confidence:** 3

**Summary:**

This paper advances multivariate time series (MTS) anomaly detection in three distinct aspects.  First, it provides theoretical insights into how the Transformer encoder represents and learns from the MTS data, revealing how its representations relate to classical time series. For instance, the authors equate the embedding process to Vector Moving Average (VMA) filtering, and the self-attention mechanism to the Space-Time Autoregressive (STAR) structure. Second, the authors propose Attention Low-Rank Transformer (ALoRa-T), which consists of the LightMTS-Embed module and Attention Low-Rank (ALoRa) layers, and a decoder. Lastly, given this new architecture, the authors propose a novel detection score and localization method: ALoRa-T score and ALoRa-Loc method.

**Strengths:**

- The authors theoretically relate the Transformer architecture back to the techniques from classical time series modeling. Based on this insight, they propose technically sound and well-motivated modifications to the Transformer architecture, further specializing it for the task of MTS anomaly detection.

- The authors propose novel detection and localization frameworks that are more reliable than previously used metrics.

- Together, the proposed method and detection/localization methods successfully outperform other baselines. The experimental results are quite comprehensive, and the authors have included code and sufficient experimental details to reproduce the results.

**Weaknesses:**

- According to Table 1, it appears that AloRa-Det is more effective on some datasets (ex) HAI or SMD) than other (SwAT, MSL). What causes such a discrepancy in the results? Is ALoRa-Det more effective at detecting certain anomaly types than others?

- The majority of the baselines are drawn from Transformer-backed anomaly detection methods (for a good reason). Yet, it would be helpful to add some baselines from other families of MTS anomaly detection methods, such as reconstruction or contrastive learning-based methods.

- Do the authors expect their method to stay functional in application scenarios where anomalies and distributional shifts (concept drifts) appear mixed together? If so, how could ALoRa-T be extended or modified to such cases?

- Although the authors present ablation studies in Section D, I believe a more thorough ablative study that investigates the effectiveness of each proposed technical component separately to assess its contribution is necessary.

- Just a minor comment on paper formatting: I understand that the authors have chosen to move many of the experimental results due to the page constraint, but I personally think key results and analyses should still remain as a part of the main manuscript. I suggest that the authors truncate some of the materials in the introduction/related works to make room for the results section.

**Questions:**

Please refer to the weaknesses above.

---

> ### Author Response · Authors · 2025-11-24
>
> We thank the Reviewer for his/her in-depth review of our work and for providing us with very insightful comments that have helped us improve the presentation and the quality of the paper.
>
>
> ## Reply to Weaknesses:
> ### Reply to Weakness 1:
> (a) To address the Reviewer’s comment, in the revised paper we have conducted a series of new experiments that evaluate the performance of the proposed ALoRa-T score across different anomaly types. The results are discussed in Appendix F, specifically in the paragraph *“Effectiveness of the ALoRa-T Score Across Anomaly Types.”* The new **Figure 10 in the revised paper** illustrates the effectiveness of the ALoRa-T score in capturing all the considered anomaly types and also highlights its general behavior.
>
> (b) We agree with the Reviewer that the proposed method achieves higher performance on certain datasets compared to others, like SWaT and MSL.
> **Why this happens**: Each dataset exhibits its own dynamics and unique spatial-temporal relationships. Consequently, it is common in the literature for different methods to perform better on certain datasets than others, depending on the complexity and inherent characteristics of each dataset.
>
> **However, for the data we studied, our method consistently outperforms all the other SOTA methods.**
>
>
> ### Reply to Weakness 2:
> To address this comment, we have included two additional state-of-the-art methods, one from each of the suggested categories.
>
> 1.  **DCdetector**, a contrastive learning–based method [R1].
>
> 2.  **DAEMON**, a reconstruction-based encoder–decoder model built upon variational autoencoders (VAE) [R2].
> **Their performance has been added to Table 1 of Section 6.2 (_Results_)**
>
> We would also like to clarify that in the submission we had already included reconstruction-based methods from different model families. For example, Transformer-based reconstruction methods, diffusion-based models such as **D3R**, and recurrent neural network–based approaches such as **OmniAnomaly** and **InterFusion**.
>
> ### Reply to Weakness 3:
> We thank the Reviewer for giving us the opportunity to clarify this point. Distribution drift is indeed an important challenge in real-world systems, and there is a dedicated research area focused specifically on this problem. Although addressing distribution drift is not the primary focus of our work, we fully acknowledge that an effective anomaly detection method should be capable of handling situations where the notion of “normality” evolves over time, requiring models to adapt accordingly.
>
> Below, we explain how our method can still remain effective under such conditions:
>
> 1. **ALoRa-T score and concept drift:**
>    The ALoRa-T score is based on changes in the ranks of self-attention values. Because the attention matrices are computed *online* as new data arrive, the score naturally adapts to concept drift. An example of this behavior is shown in the new **Figure 10** in Appendix F of the revised version of our paper. Specifically, we refer to the **trend anomaly type**, which the ALoRa-T score successfully detects. At the same time, this trend shift introduces **distribution drift** (in particular, a change in the mean value of the time series). However, since the underlying system dynamics remain unchanged, ALoRa-T does not interpret this drift as an anomaly. Instead, it correctly treats it as part of the evolving normal behavior.
> 2.  **Model adaptation for reconstruction:**
>     The reconstruction component of ALoRa-Det method, would need to be integrated into a framework that supports incremental or continual learning in order to adapt to new normal patterns over time. A sliding-window strategy could be used to determine when retraining or partial updates are necessary, for example, when the system detects that the underlying normality has changed.
>
> In Section 8 (Conclusion) of the revised paper, we have included this as a direction for further future work.

---

> ### Author Response · Authors · 2025-11-24
>
> ### Reply to Weakness 4:
>
> We thank the Reviewer for this comment, which helped us provide additional information to support further our work. **In the revised version**, we have included the following ablation studies:
>
> **Main paper, Section 7 of the revised paper ( _Ablation studies_)**
> 1. **Ablation study on the effectiveness of ALoRa-Loss and ALoRa-Score:** Table 3 of the revised paper.
>
> **Conclusion**: The **results** demonstrate the **effectiveness of both the ALoRA loss and the ALoRA-T score** in improving anomaly detection across all datasets. This highlights the robustness and general applicability of these components.
>
> 2. **Extended ablation study on the LightMTS-Embed module**: We added **Table 4**  in Section 7, which reports the selection of Top-K pairs versus using all pairs, and compares Spearman correlation with Pearson correlation.
>
> **Conclusion**: Selecting the Top-K pairs substantially reduces computational cost without sacrificing accuracy, and even improves it in some cases, demonstrating the effectiveness of this design.
>
>
> **Appendix D of the revised paper (_Ablation studies supplement_)**
>
> 3. **Ablation Study on the Top-K pairs selection: The new Table 8** illustrates the sensitivity of the parameter $K$ for the LightMTS-Embed module.
>
> **Conclusion**: The results indicate that the model is not sensitive to this choice.  Since our model is already the most computationally efficient, as demonstrated in the original paper (Appendix C), and its computational cost is already very low, making it suitable for real-world applications, further reducing $K$ was unnecessary.
>
> 4. **Ablation study supporting the decision to omit feed-forward layers**:
> In the original paper, the decision to omit feed-forward layers was theoretically motivated by Proposition 1 (statement 3), which shows that these layers do not alter the latent space structure. To further support this claim, we have added an ablation study evaluating both performance and computational cost when feed-forward layers are used and when they are omitted. **This study is presented in Appendix D.2  and summarized in Table 9 of the revised paper**.
>
> **Conclusion**: The experimental results in Table 9 confirm the theoretical insight. Specifically, when feed-forward layers are included, performance either decreases or remains unchanged, while computational cost increases substantially.
>
> 5. **Ablation study on the placement of the ALoRa-Loss**, comparing head-wise application versus averaging across self-attention heads;
> See  **Appendix D.3** and **Table 10** of the revised paper.
>
> **Conclusion**: Both variants achieve nearly identical detection performance across all datasets. However, applying the ALoRA loss to the average self-attention matrix is slightly more computationally efficient, and since there is no other advantage to the alternative, this variant was preferred.
>
> **Reply to Weakness 5:**
> We thank the Reviewer for acknowledging that page limitations were the primary reason some experimental results were placed in the appendix. **With the additional page allowed in the revised version**, we have added an additional paragraph on ablation studies to the main paper (Section 7). We remain open to any specific suggestions regarding further adjustments to the balance between results presented in the main paper and those included in the appendix.
>
>
> ## References
> [R1] Yang et al. , Dcdetector: Dual attention contrastive representation learning for time series anomaly detection, KDD, 2023.
>
> [R2] Chen et al. , Adversarial autoencoder for unsupervised time series anomaly detection and interpretation, WSDM, 2023.

---

### Meta-Review · Area_Chair_SyA3 · 2026-01-04

**Summary:**

Main concerns included various clarifications and reasonable additional small experiments (e.g., ablation studies), which were addressed by the authors during the rebuttal phase. Considering that generally the reviewers were positive and that the one more negative review requested to increase their score, I believe overall the paper worths a spot at the conference. I recommend acceptance as poster.

**Reviewer Concerns:**

All clarifications were addressed and authors provided new experiments to address various concerns. No major concerns are outstanding

**Reviewer Scores:**

Overall reviews were positive and the one more negative review requested to increase their score.

---

### Decision · Program_Chairs · 2026-01-26

Accept (Poster)